# Learning a Cross-Modal Schrödinger Bridge for Visual Domain Generalization

**Hao Zheng**[1]†, **Jingjun Yi**[2,3]†, **Qi Bi**[4✉], **Huimin Huang**[1], **Haolan Zhan**[5], **Yawen Huang**[1],
**Yuexiang Li**[6✉], **Xian Wu**[1], **Yefeng Zheng**[2✉]

†: equal contribution
[1]Tencent Jarvis Lab, China, [2]Westlake University, China
[3]University of Alberta, Canada, [4]University of Amsterdam, the Netherland
[5]Monash University, Australia, [6]University of Macau, Macau
q.bi@ieee.org, yuexiang.li@ieee.org
zhengyefeng@westlake.edu.cn

## Abstract

Domain generalization aims to train models that perform robustly on unseen target domains without access to target data. The realm of vision-language foundation model has opened a new venue owing to its inherent out-of-distribution generalization capability. However, the static alignment to class-level textual anchors remains insufficient to handle the dramatic distribution discrepancy from diverse domain-specific visual features. In this work, we propose a novel cross-domain Schrödinger Bridge (SB) method, namely SBGen, to handle this challenge, which explicitly formulates the stochastic semantic evolution, to gain better generalization to unseen domains. Technically, the proposed `SBGen` consists of three key components: (1) *text-guided domain-aware feature selection* to isolate semantically aligned image tokens; (2) *stochastic cross-domain evolution* to simulate the SB dynamics via a learnable time-conditioned drift; and (3) *stochastic domain-agnostic interpolation* to construct semantically grounded feature trajectories. Empirically, `SBGen` achieves state-of-the-art performance on domain generalization in both classification and segmentation. This work highlights the importance of modeling domain shifts as structured stochastic processes grounded in semantic alignment.

## 1  Introduction

Distribution shift is a fundamental challenge in both machine learning and computer vision. Domain Generalization (DG) addresses this challenge by training models on one or more source domains that can generalize well to unseen target domains [45, 31, 75, 16]. A generalizable representation is especially critical for trust-worthy artificial intelligence and plays a pivot role in safety-crucial applications, such as autonomous driving [32, 74, 8, 77, 21] and medical imaging [11, 68, 7, 76], where target environments are not available during training.

In visual domain generalization, the images from various domains are usually diverse in terms of the contrast, texture, illumination, and resolution [88, 59, 48]. The emergence of vision-language models (VLM) [59] has opened up a new venue to approach the DG problem. Its general idea is that, the category-wise text description is capable to anchor high-level semantics despite the distribution shift of the images from various unseen domains [31]. Specifically, existing VLM based DG methods usually treat the VLM as a static feature extractor and apply fixed alignment strategies such as prompt learning [86], cosine matching [1] and adversarial regularization [81] to enforce similarity between image features and class-level text queries.

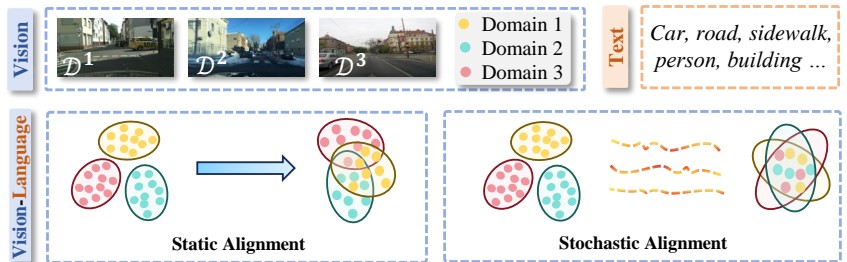

Figure 1: Leveraging domain-agnostic category-wise text embedding to align the domain-specific visual features from different domains is a common paradigm for domain generalization in vision-language models (VLM). **Left**: Existing methods usually rely on a static alignment strategy, which can be insufficient to handle the dramatic discrepancy and spurious correlations across domains. **Right**: In contrast, this paper presents `SBGen`, a stochastic alignment strategy to enhance the domain generalization ability of VLM.

Crucially, such fixed alignment strategies are developed under the assumption that visual embeddings can be directly and reliably projected onto the textual anchors, which may not necessarily hold under the domain shift. In fact, visual features extracted from unseen domains by an image encoder may exhibit dramatic distribution discrepancy (shown in Fig. 1). Such degraded semantics and spurious correlations may not be corrected by static alignment. Moreover, such a deterministic projection may not offer a clear path to model the semantic drift caused by the domain shift, as it can be difficult to map onto the textual semantics in a single step. Conversely, if we turn to the multi-step based mapping, how to transform the partially-aligned intermediate states to the semantic invariance is the key challenge.

This paper pushes this frontier by approaching VLM based DG from a fundamentally different perspective. Rather than enforcing the static similarity between image and text features, we ask, *what if model their alignment as a stochastic semantic evolution? Is it possible to gain better generalization to unseen domains?*

We propose `SBGen`, a novel cross-modal Schrödinger bridge for visual domain generalization, to realize the above objectives. Its general idea is to formulate the alignment from domain-specific visual representations to domain-agnostic textual semantics, as a controlled stochastic process that interpolates between two distributions while remaining close to a prior. It comprises three main stages. First, a *text-guided domain-aware feature selection* component is proposed to extract local visual tokens from source images that align with class-level textual queries, which focuses the model on semantically relevant content while avoiding domain-specific interruption. Next, a *stochastic cross-domain evolution* component is proposed to model the Schrödinger Bridge as a time-indexed stochastic differential equation (SDE) with a learnable, query-conditioned drift. This process is then discretized and simulated to generate a trajectory of evolving features. Finally, a *stochastic domain-agnostic interpolation* component is proposed, using these features to bridge source representations with semantic anchors.

Notably, each stage in the proposed `SBGen` is differentiable and can be jointly trained with a loss function that balances task supervision and stochastic consistency. It enables us to simulate a sequence of latent feature states that progressively reduce domain-specific bias and converge toward semantic consistency with textual queries. In contrast to prior methods, our approach supports structured semantic interpolation, and models an interpretable and probabilistic trajectory from biased source features toward text-grounded, domain-agnostic representations. The proposed `SBGen` is evaluated on standard domain generalization benchmarks for both classification and segmentation. It consistently outperforms the state-of-the-art methods.

Concretely, our contributions can be summarized as follows.

- We propose cross-modal Schrödinger Bridge for visual domain Generalization (`SBGen`), a novel framework that aligns domain-specific image features with domain-agnostic textual semantics via Schrödinger Bridge–guided stochastic evolution.

- We introduce a principled three-stage pipeline that performs text-guided feature selection, stochastic cross-domain evolution, and semantically anchored interpolation.

- We provide a theoretical justification of the proposed SBGen through a provably tighter generalization bound, and demonstrate its effectiveness on multiple DG benchmarks in both classification and segmentation settings.

## 2 Related work

**Vision-Language Models (VLMs)** have emerged as effective tools for capturing deep semantic relationships across modalities. Some typical VLMs include CLIP [59], ALIGN [34] and EVA02 [26, 25]. The expressive representations have proven effective for more complex downstream vision-language tasks [40, 44, 87].

**Domain Generalization (DG)** has been extensively studied in the machine learning community [7, 12, 73, 46, 10]. More recently, the emergence of vision-language model (VLM) [59, 86] has paved a new path for DG. One research line focuses on leveraging its inherent out-of-distribution generalization ability [1, 23, 42]. Another closer research line usually leverages the text embedding to augment the domain diversity or to statically align the domain-specific visual features [12, 3, 50, 13, 43, 82, 65, 16, 36, 39, 15, 79]. However, the majority of these approaches rely on the assumption that domain-specific visual features can be directly and statically matched to domain-agnostic textual embeddings, which may overlook the dynamic and multi-faceted characteristics of semantic shifts caused by domain changes.

**Domain Generalized Semantic Segmentation (DGSS)** aims to learn a generalizable segmentation model trained only on a source domain. Earlier works usually use normalization [53, 54], whitening [17, 57] or mask attention [21, 9]. Other works use style hallucination or randomization techniques for domain augmentation [38, 83, 84, 35, 80]. More recently, vision foundation models (VFM) [74, 77, 8] and VLMs [22, 32] have been used for DGSS. Despite these advancements, these approaches usually implement a direct and static alignment between the image and text embeddings, or enrich the domain diversity guided by the text description. They may still be insufficient to handle the semantic drift caused by the dramatic domain shift.

**Schrödinger Bridge and Stochastic Feature Transport** [20, 70] have drawn increasing attention. These frameworks define stochastic processes that interpolate between distributions via entropy-regularized optimal transport. Recent work has explored SBs for generative modeling [72, 55, 64], score-based dynamics [19, 66, 69], depth estimation [28], and modality translation [68, 4, 33]. However, how to explicitly model the stochastic alignment from domain-specific features to domain-agnostic semantics using SB-driven dynamics for DG remains underexplored.

## 3 Preliminaries

**Problem Definition.** Let $\mathcal{X}$ and $\mathcal{Y}$ denote the space of input images and the space of structured labels from a certain task (*e.g.*, classification). Given a set of labeled source domains $\mathcal{D}^S = \{(x_n^S, y_n^S)\}_{n=1}^{N_S}$ with $x_n^S \in \mathcal{X}, y_n^S \in \mathcal{Y}$, and a set of unseen target domains $\mathcal{D}^U = \{(x_m^U, y_m^U)\}_{m=1}^{N_U}$, the objective is to train a model on source domains that generalizes to these unseen domains.

**Definition 1. Optimal Transport (OT).** *Let $P^S$ and $P^U$ be two probability distributions over $\mathbb{R}^C$, respectively. The classical OT problem seeks a deterministic transport map $M : \mathbb{R}^C \to \mathbb{R}^C$ minimizing a transport cost:*

$$\min_{M:M_\# P^S = P^U} \mathbb{E}_{z^S \sim P^S} \left[ \|\boldsymbol{z}^S - M(\boldsymbol{z}^S)\|^2 \right], \tag{1}$$

*where $M_\# P^S$ denotes the pushforward measure of $P^S$ through $M$.*

**Definition 2. Entropy-Regularized OT.** *A stochastic coupling $\pi(\boldsymbol{z}^S, \boldsymbol{z}^U)$ with marginal constraints $\pi \in \Pi(P^S, P^U)$ is introduced to improve the robustness of OT, minimizing:*

$$\min_{\pi \in \Pi(P^S, P^U)} \int \|\boldsymbol{z}^S - \boldsymbol{z}^U\|^2 \, d\pi(\boldsymbol{z}^S, \boldsymbol{z}^U) + \varepsilon \cdot \mathrm{KL}(\pi \| \mathcal{R}), \tag{2}$$

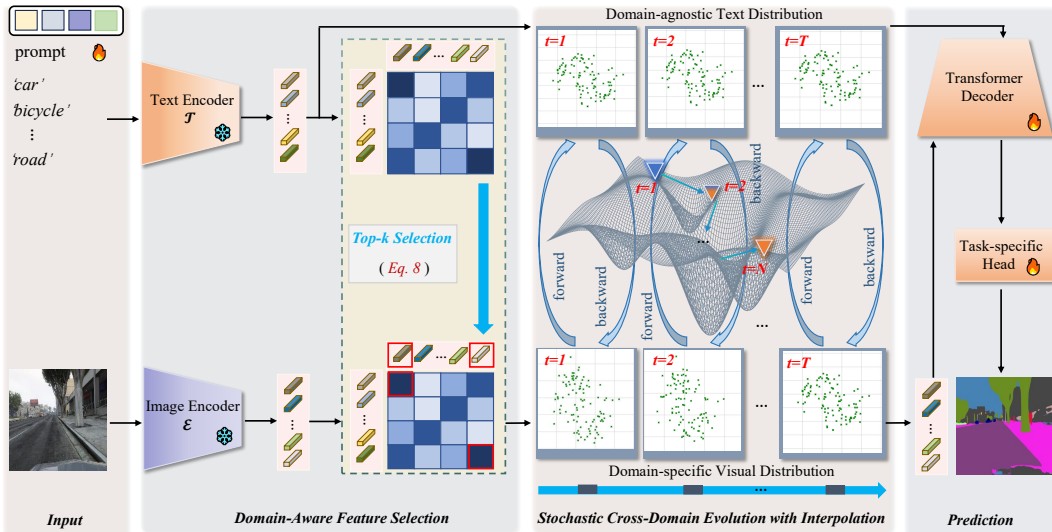

Figure 2: **Overall pipeline of** SBGen. Step 1: We generate initial textual object queries $\mathbf{q}_t^0$ from the $K$ class text embeddings $\{\mathbf{t}_k\}_{k=1}^K$. Step 2: To improve the segmentation capabilities of these queries, we incorporate text-to-pixel attention within the pixel decoder. This process enhances the semantic clarity of pixel features, while reconstructing high-resolution per-pixel embeddings $\mathbf{Z}$. Step 3: The transformer decoder refines these queries for the final prediction. Each prediction output is then assigned to its corresponding ground truth through fixed matching, ensuring that each query consistently represents the semantic information of one class.

*where $\mathcal{R}$ is a reference measure and $\varepsilon > 0$ controls the regularization strength, enabling stochastic transport but lacks a notion of dynamics over time.*

**Definition 3. Schrödinger Bridge (SB).** *OT is extended to the dynamic setting by introducing a continuous-time stochastic process $\{P_t\}_{t \in [0,1]}$ that evolves from $P^S$ to $P^U$, while being minimally deviated from a prior diffusion process $\mathbb{P}$ (e.g., Brownian motion). The SB formulation is:*

$$\min_{\mathbb{Q}} \mathrm{KL}(\mathbb{Q} \| \mathbb{P}) \quad \text{subject to} \quad \mathbb{Q}_{t=0} = P^S, \quad \mathbb{Q}_{t=1} = P^U, \tag{3}$$

*where $\mathbb{Q}$ denotes the law of the interpolating process over latent features. This yields a family of time-indexed distributions $P_t$ modeling the optimal evolution of visual features across domains.*

## 4 Methodology

We propose SBGen, a cross-modal Schrödinger Bridge framework for visual domain Generalization. Its general idea is to learn a time-indexed stochastic process over feature distributions, evolving from the source domain toward target-aligned representations, guided by the domain-agnostic class-level textual queries. Specifically, it consists of three components, namely, Domain-aware Visual Feature Selection, Stochastic Cross-Domain Evolution, and Stochastic Domain-Agnostic Interpolation. The rigorous generalization error analysis on its upper bound is provided the supplementary material.

### 4.1 Domain-aware Visual Feature Selection

Assume we have an image encoder $\mathcal{E}$ (e.g., CLIP-ViT), which extracts the visual features from an image $x \in \mathcal{X}$, given by $\mathcal{F} = \mathcal{E}(x) \in \mathbb{R}^{H \times W \times C}$. We also assume access to a text encoder $\mathcal{T}$ and a set of class names $\mathcal{Q}_c$ ($c = 1, \cdots, N_c$), where $N_c$ is the number of semantic classes. Following the textual query generation protocol [52], each class name is converted into a class-specific textual query and embedded via $\mathcal{T}$, resulting in $\boldsymbol{q}_c = \mathcal{T}(\mathcal{Q}_c) \in \mathbb{R}^{1 \times C}$. The visual features $\mathcal{F}$ include information such as background, textures, lighting, or object co-occurrence, which can be usually domain-specific. In contrast, the class-specific textual embedding $\boldsymbol{q}_c$ tends to capture high-level and domain-invariant semantics. To leverage the class-wise text embeddings $\boldsymbol{q}_c$ as domain-agnostic anchors for semantic alignment, the domain-aware visual feature selection component is proposed.

Rather than using all visual tokens uniformly, we identify class-conditioned regions in the image feature space that exhibit high alignment with textual descriptions. This targeted selection filters out the irrelevant content and yields a semantically grounded set of features for downstream modeling. Specifically, the visual feature $\mathcal{F} \in \mathbb{R}^{H \times W \times C}$ can be regarded as a set of feature vectors $\{\mathcal{F}_{h,w} \in \mathbb{R}^{1 \times C}\}$, and each $\mathcal{F}_{h,w}$ corresponds to the representation of a certain spatial position $(h, w)$ in the image feature. Then, the cosine similarity between each spatial feature $\mathcal{F}_{h,w}$ and each class embedding $\boldsymbol{q}_c$ is computed as

$$\mathcal{S}_{h,w,c} = \left\langle \frac{\mathcal{F}_{h,w}}{\|\mathcal{F}_{h,w}\|}, \frac{\boldsymbol{q}_c}{\|\boldsymbol{q}_c\|} \right\rangle, \quad \forall h, w, c. \tag{4}$$

Then, the top-$k$ spatial locations for each class $c$ that exhibit the highest alignment scores are selected, which distinguishes the semantically relevant and domain-robust feature locations, avoiding noisy or background-dominated inputs [60, 89]. Let $\{\boldsymbol{z}_0^{(i)}\}_{i=1}^N \subset \mathbb{R}^{1 \times C}$ denote the corresponding set of visual tokens, which serve as the class- and domain-aware initial feature set. These empirical samples from the source distribution $P_0$ are used as input to our Schrödinger Bridge evolution process.

## 4.2 Stochastic Cross-Domain Evolution

Most prior VLM based DG methods adopt deterministic feature mappings or prompt tuning strategies, which are limited in capturing uncertainty or adapting to structured variation between domains. We explicitly model the evolution from the domain-specific visual features to the domain-agnostic text embeddings, so as to enhance the generalization to unseen target domains. To realize this objective, this evolution is modeled as a stochastic process governed by a Schrödinger Bridge.

The selected features $\boldsymbol{z}_0$ serve as the initial samples to form the empirical source distribution $\boldsymbol{z}_0 \sim P^S$. Specifically, we define a time-indexed stochastic process $\{\boldsymbol{z}_t\}_{t \in [0,1]} \subset \mathbb{R}^C$ governed by the following stochastic differential equation (SDE), given by

$$\mathrm{d}\boldsymbol{z}_t = f_\theta(\boldsymbol{z}_t, t)\,\mathrm{d}t + \sqrt{2\varepsilon}\,\mathrm{d}W_t, \tag{5}$$

where $f_\theta : \mathbb{R}^C \times [0,1] \to \mathbb{R}^C$ is a learnable drift function, $\varepsilon > 0$ is a fixed diffusion coefficient, and $W_t$ denotes standard Brownian motion. The process begins at $\boldsymbol{z}_0 \sim P^S$ and is regularized to terminate near a distribution $P^U$ implicitly defined by the textual query embeddings $\{\boldsymbol{q}_c\}$.

Following best practices in recent Schrödinger Bridge literature [20, 70], we parameterize the drift function $f_\theta$ using a Multi-Layer Perceptron (MLP) conditioned on both the time index and class semantics. This drift parameterization is computed as

$$f_\theta(\boldsymbol{z}_t, t) = \mathrm{MLP}_\theta\left(\mathrm{LayerNorm}(\boldsymbol{z}_t + \gamma(t) + \boldsymbol{q}_c)\right), \tag{6}$$

where $\gamma(t) \in \mathbb{R}^C$ is a sinusoidal time embedding and $\boldsymbol{q}_c$ is the target text embedding for class $c$. This design encourages smooth and semantically aligned evolution under time-aware control.

Concretely, the proposed stochastic cross-domain evolution allows the alignment between the domain-specific visual features and the semantic anchors defined by domain-agnostic text embeddings, while maintaining flexibility to domain shifts.

## 4.3 Stochastic Domain-Agnostic Interpolation

Nevertheless, the proposed stochastic cross-domain evolution only specifies the initial state from the domain-specific visual features from the source domain and the end state from the domain-agnostic text embeddings, which does not take the transitional dynamics between domain-specific representations and domain-invariant semantics. To address this issue, in the proposed SBGen, a stochastic interpolation mechanism is introduced. Its general idea is to model a continuous-time evolution of feature states that gradually transforms source-domain features into semantically aligned, domain-agnostic representations.

By simulating the Schrödinger Bridge dynamics from $t = 0$ to $t = 1$, we generate a trajectory $\{\boldsymbol{z}_t\}$ that smoothly interpolates between a domain-biased initial state from the source domain and a domain-agnostic text embeddings. Unlike deterministic mappings, this stochastic evolution accounts for uncertainty and allows for explicit control over the extent of semantic alignment. Each intermediate

feature $z_t$ can be interpreted as a partial semantic abstraction, providing generalization to unseen domains and offering flexibility in selecting the optimal representational point. Specifically, we numerically simulate the SDE in Eq. (5) using Euler–Maruyama discretization. Given a step size $\Delta t = 1/T$ and $t_n = n \cdot \Delta t$, the discretized evolution can be computed as

$$z_{t_{n+1}} = z_{t_n} + f_\theta(z_{t_n}, t_n)\,\Delta t + \sqrt{2\varepsilon\Delta t}\,\xi_n, \quad \xi_n \sim \mathcal{N}(0, I), \tag{7}$$

with initial condition $z_{t_0} = z_0 \sim P_0$. The sequence $\{z_t\}$ captures the full evolution. We implement this via Monte Carlo estimation over minibatches and simulate each path using Eq. (7).

We adopt this discretization to efficiently simulate the evolution from domain-specific features $z_0$ to domain-invariant features $z_T$ through a sequence of stochastic updates. The terminal feature $z_{t_T}$ is given by the unrolled summation:

$$z_{t_T} = z_{t_0} + \sum_{n=0}^{T-1} f_\theta(z_{t_n}, t_n)\,\Delta t + \sum_{n=0}^{T-1} \sqrt{2\varepsilon\Delta t}\,\xi_n, \tag{8}$$

which represents a stochastic interpolation path toward the domain-agnostic class-wise text semantics.

### 4.4  Prediction, Optimization & Implementation Details

The evolved features $z_T \in \mathbb{R}^{K \times C}$, obtained through the SB trajectory, are not directly used for task prediction. Instead, they serve as refined class-aware feature anchors that are written back to the original visual feature map $\mathcal{F} \in \mathbb{R}^{C \times H \times W}$. For each class $c$, the corresponding evolved embeddings $z_T^{(c)}$ are broadcasted to their original locations. This update process yields an enhanced image feature map $\mathcal{F}'$, in which the selected regions are aligned toward the class-conditional textual semantics.

The updated visual feature map $\mathcal{F}'$ and the class queries $\{q_c\}$ are then fed into the decoder $\mathcal{D}$ for the final task prediction (e.g., classification and segmentation) on unseen target domains, where classification uses global pooling followed by linear projection, and segmentation employs per-pixel decoding via cosine similarity with class embeddings.

Afterwards, the learning objective is to minimize the expected total loss over the data distribution and simulated paths, given by

$$\min_\theta\ \mathbb{E}_{x \sim \mathcal{D}_S}\ \mathbb{E}_{z_0 \sim P_0}\left[\mathcal{L}_{\text{sup}}(\mathcal{D}(\mathcal{F}', \{q_c\}), y)\ +\ \lambda \cdot \text{KL}(\mathbb{Q}_\theta \,\|\, \mathbb{P})\right], \tag{9}$$

where $\mathcal{L}_{\text{sup}}$ denotes the task-specific loss, $\mathbb{Q}_\theta$ denotes the forward path distribution induced by the learned drift $f_\theta$ to transport initial visual features $z_0 \sim P_0$ toward class-conditional textual anchors $q_t$, $\mathbb{P}$ denotes the reference Brownian motion starting at $z_0$, and $\lambda \in \mathbb{R}_{>0}$ is a regularization weight. The KL loss $\text{KL}(\mathbb{Q}_\theta \,\|\, \mathbb{P})$ is approximated by $\sum_{i=0}^{T-1} \|f_\theta(z_{t_i}, t_i)\|^2 \Delta t/4\varepsilon + \|z_{t_T} - q_c\|^2$ for each pair of $z_0$ and the corresponding $q_c$.

For the classification task, the image encoder $\mathcal{E}$ and the text encoder $\mathcal{T}$ use the pre-trained CLIP in align with the prior DG methods. The task-specific decoder $\mathcal{D}$ is a linear layer followed by a Softmax layer. For the segmentation task, following prior domain generalized semantic segmentation methods [52, 67], the image encoder $\mathcal{E}$ and the text encoder $\mathcal{T}$ use the pre-trained EVA-02 [25]. The task-specific decoder $\mathcal{D}$ integrates the pixel decoder of the Mask2Former model [14]. The hyperparameters and configurations of both tasks are detailed in the supplementary material.

## 5  Experiments

### 5.1  Results on Domain Generalization in Classification

**Datasets & Evaluation Metrics.** PACS [41], VLCS [24], OfficeHome [71], TerraIncognita [5], and DomainNet [58] comprise of 9,991, 10,729, 15,588, 24,330 and 0.6 million images from four, four, four and six domains, respectively. In line with prior work [29, 12], the leave-one-domain-out evaluation protocol is adopted, where one domain is held out as the unseen target domain, while the remaining domains are used for training the model. Performance is reported using classification accuracy (percentage, %) as the evaluation metric.

**Compared Methods.** Existing VLM based domain generalization methods are involved for comparison, namely, SWAD [12], CLIP [59], SMA [3], DUPRG [50], CoOp [86], MIRO [13], SEDGE

Table 1: Comparison with the state-of-the-art methods on PACS, VLCS, OfficeHome, DomainNet and TerraInc. By default the results are cited from [15, 65, 16, 36, 79]. Evaluation metric is classification accuracy (in %). Top three results are highlighted as best , second and third , respectively.

| Method | Venue | PACS | VLCS | OfficeHome | DomainNet | TerraInc | Avg |
|---|---|---|---|---|---|---|---|
| *ResNet-50 Pre-trained by ImageNet:* | | | | | | | |
| DANN [27] | IJCAI'2016 | 83.6 | 78.6 | 65.9 | 38.3 | 46.4 | 65.6 |
| Fish [63] | ICML'2022 | 85.5 | 77.8 | 68.6 | 42.7 | 45.1 | 63.9 |
| DAC-SC [37] | CVPR'23 | 87.5 | 78.7 | 70.3 | 44.9 | 46.5 | 65.6 |
| SAGM [73] | CVPR'2023 | 86.6 | 80.0 | 70.1 | 45.0 | 48.8 | 66.1 |
| *ViT-B/16 Pre-trained by CLIP:* | | | | | | | |
| SWAD [12] | NIPS'2021 | 91.3 | 79.4 | 76.9 | 51.7 | 45.4 | 68.9 |
| CLIP [59] | ICML'2021 | 96.2 | 81.7 | 82.0 | 57.5 | 33.4 | 70.2 |
| SMA [3] | NIPS'2022 | 92.1 | 79.7 | 78.1 | 55.9 | 48.3 | 70.8 |
| DUPRG [50] | ICLR'2023 | 97.1 | 83.9 | 83.6 | 59.6 | 42.0 | 73.2 |
| CoOp [86] | IJCV'2022 | 96.2 | 77.6 | 83.9 | 59.8 | 48.8 | 73.3 |
| MIRO [13] | ECCV'2022 | 95.6 | 82.2 | 82.5 | 54.0 | 54.3 | 73.7 |
| SEDGE [43] | ArXiv'2022 | 96.1 | 82.2 | 80.7 | 54.7 | 56.8 | 74.1 |
| DPL [82] | TAI'2023 | 97.3 | 84.3 | 84.2 | 56.7 | 52.6 | 75.0 |
| CLIPOOD [65] | ICML'2023 | 97.3 | 85.0 | 87.0 | 63.5 | 60.4 | 78.6 |
| Promptstyler [16] | ICCV'2023 | 97.2 | 82.9 | 83.6 | 59.4 | - | - |
| KAdaptaion [36] | WACV'2025 | **97.5** | 83.0 | **90.3** | 62.7 | 51.9 | 77.1 |
| GESTUR [39] | ICCV'2023 | 96.0 | 82.8 | 84.2 | 58.9 | 55.7 | 75.5 |
| DPR [15] | CVPR'2024 | **97.5** | 86.4 | 86.1 | 62.1 | 57.1 | 77.8 |
| CLIPCEIL++ [79] | NeurIPS'2024 | 97.2 | 85.2 | 87.7 | 63.6 | 62.0 | 79.1 |
| Ours | 2025 | 97.4 | **86.7** | 89.9 | **64.4** | **63.5** | **80.4** |

[43], DPL [82], CLIPOOD [65], Promptstyler [16], KAdaptaion [36], GESTUR [39], DPR [15] and CLIPCEIL++ [79]. Several prior ImageNet pre-trained domain generalization methods, namely, DANN [27], Fish [63], DAC-SC [37] and SAGM [73], are also compared for boarder reference.

**Results.** Table 1 reports the outcomes on the five datasets. The proposed method shows the state-of-the-art performance over the existing VLM based DG methods, yielding a classification accuracy of 86.7%, 64.4%, and 63.5% on VLCS, DomainNet and TerraInc, respectively. Notably, DomainNet and TerraInc are particularly large-scale, indicating the scalability of the proposed method. Its performance is also very close to the state-of-the-art on PACS and VLCS, where both benchmarks have been highly saturated. Overall, the proposed method shows the best performance on the average accuracy of five datasets, outperforming the second-best by 1.3%.

## 5.2 Results on Domain Generalized Semantic Segmentation

**Datasets & Evaluation Metrics.** Four driving-scene semantic segmentation datasets that share 19 common scene categories are used for validation. Specifically, **CityScapes** (C) [18] consists of 2,975 and 500 images for training and validation, respectively. The images were captured under the clear conditions in tens of Germany cities. **BDD-100K** (B) [78] has 7,000 and 1,000 images for training and validation, respectively. The images were captured under diverse conditions from a variety of global cities. **Mapillary** (M) [47] is a large-scale semantic segmentation dataset, which consists of 25,000 images from diverse conditions. **GTA5** (G) [61] is another synthetic dataset, which has 24,966 simulated images from the American street landscape. Following the evaluation protocol of existing foundation model based DGSS methods [74, 52], two commonly-used evaluation settings are: 1) G → C, B, M; and 2) C → B, M, respectively. The evaluation metric is mean Intersection of Union (mIoU, in percentage %). All the experiments report the average outcomes from three independent repetitions.

**Compared Methods.** We compare with existing DGSS methods from three major categories: 1) ResNet based methods, namely, ISW [17], GTR [56], SHADE [83], SAW [57], WildNet [38], AdvStyle [85], SPC [30], and BlindNet [2]; 2) Mask2Former based methods, namely, HGFormer [21] and CMFormer [9]; 3) VFM and VLM based methods, namely, DIDEX [49], REIN [74], SET [77], FADA [8], tqdm [52], and MGRNet [67]. By default, the performance is directly cited from prior works [9, 49, 74, 8, 52, 67], and we report two decimal results. '*' denotes that the original paper only reported one decimal results.

Table 2: Performance comparison between the proposed method and existing DGSS methods. C: CityScapes [18]; B: BDD-100K [78]; M: Mapillary [47]; S: SYNTHIA [62]; G: GTA5 [61]. '-': results were not reported and official source code is not available; '*': only reported one decimal official results; '†': re-implementation with official source code under default settings. Evaluation metric is mIoU in %. Top three results are highlighted as best , second and third , respectively.

| Method | Venue | Encoder | G → C | G → B | G → M | Avg. | C → B | C → M | Avg. |
|---|---|---|---|---|---|---|---|---|---|
| *ImageNet Pretrained:* | | | | | | | | | |
| ISW [17] | CVPR'2021 | ResNet-101 | 36.58 | 35.20 | 40.33 | - | 50.73 | 58.64 | - |
| GTR [56] | TIP'2021 | ResNet-101 | 37.53 | 33.75 | 34.52 | - | 50.75 | 57.16 | - |
| SHADE [83] | ECCV'2022 | ResNet-101 | 44.65 | 39.28 | 43.34 | - | 50.95 | 60.67 | - |
| SAW [57] | CVPR'2022 | ResNet-101 | 39.75 | 37.34 | 41.86 | - | 52.95 | 59.81 | - |
| WildNet [38] | CVPR'2022 | ResNet-101 | 44.62 | 38.42 | 46.09 | - | 50.94 | 58.79 | - |
| AdvStyle [85] | NeurIPS'2022 | ResNet-101 | 39.62 | 35.54 | 37.00 | - | - | - | - |
| SPC [30] | CVPR'2023 | ResNet-101 | 44.10 | 40.46 | 45.51 | - | - | - | - |
| BlindNet [2] | CVPR'2024 | ResNet-101 | 45.72 | 41.32 | 47.08 | - | 51.84 | 60.18 | - |
| HGFormer*[21] | CVPR'2023 | Swin-T | - | - | - | - | 53.4 | 66.9 | - |
| CMFormer [9] | AAAI'2024 | Swin-B | 55.31 | 49.91 | 60.09 | - | 59.27 | 71.10 | - |
| *VLM Pretrained:* | | | | | | | | | |
| DIDEX*[49] | WACV'2024 | Stable Diffusion | 62.0 | 54.3 | 63.0 | 59.7 | - | - | |
| VLTSeg*[32] | ACCV'2024 | CLIP-L | 55.6 | 52.7 | 59.6 | 56.0 | - | - | - |
| REIN*[74] | CVPR'2024 | EVA02-L | 65.3 | 60.5 | 64.9 | 63.6 | 64.1 | 69.5 | 66.8 |
| SET*[77] | MM'2024 | EVA02-L | 66.4 | 61.8 | 65.6 | 64.6 | - | - | - |
| FADA*[8] | NeurIPS'2024 | EVA02-L | 66.7 | 61.9 | 66.1 | 64.9 | - | - | - |
| tqdm [52] | ECCV'2024 | EVA02-L | 68.88 | 59.18 | 70.10 | 66.05 | 64.72 | 76.15 | 70.44 |
| MGRNet [67] | AAAI'2025 | EVA02-L | 69.53 | 61.14 | 69.97 | 66.88 | 64.70 | 76.43 | 70.56 |
| Ours | | EVA02-L | **71.24** | **62.26** | **71.91** | **68.74** | **66.03** | **77.90** | **71.97** |
| | | | ↑1.71 | ↑1.12 | ↑1.94 | ↑1.59 | ↑1.33 | ↑1.47 | ↑1.41 |

Table 3: Generalization test on various vision-language models. '*': only reported one decimal official results.

| Method | DINOv2[51] | | | | CLIP[59] | | | |
|---|---|---|---|---|---|---|---|---|
| | G → C | G → B | G → M | Avg. | G → C | G → B | G → M | Avg. |
| REIN*[74] | 66.4 | 60.4 | 66.1 | 64.3 | 57.1 | 54.7 | 60.5 | 57.4 |
| SET [77] | 68.06 | 61.64 | 67.68 | 65.79 | 58.2* | 55.3* | 61.4* | 58.3* |
| FADA [8] | 68.23 | 61.94 | 68.09 | 66.09 | 58.7* | 55.8* | 62.1* | 58.9* |
| MGRNet [67] | **73.87** | 62.91 | 73.52 | 70.10 | 62.31 | 56.09 | 66.47 | 61.62 |
| Ours | 72.85 | **63.59** | **73.90** | **70.11** | **63.17** | **57.82** | **66.94** | **62.64** |
| | ↓-1.02 | ↑0.68 | ↑0.38 | ↑0.01 | ↑0.86 | ↑1.73 | ↑0.47 | ↑1.02 |

Table 4: Comparison between stochastic evolution and static alignment methods.

| Method | G→C | G→B | G→G | Avg. |
|---|---|---|---|---|
| Baseline | 68.88 | 59.18 | 70.10 | 66.05 |
| DCM | 69.78 | 60.92 | 70.84 | 67.18 |
| w.o. TID | 70.01 | 61.16 | 71.13 | 67.43 |
| Ours | **71.24** | **62.26** | **71.91** | **68.74** |

Table 5: Ablation studies on each component of the proposed method. Evaluation metric is mIoU in %.

| | Component | G → C | G → B | G → M | Avg. | C → B | C → M | Avg. |
|---|---|---|---|---|---|---|---|---|
| 1) | Baseline | 68.88 | 59.18 | 70.10 | 66.05 | 64.72 | 76.15 | 70.44 |
| 2) | DFS | 69.45 | 60.07 | 70.04 | 66.52 | 65.17 | 76.85 | 71.01 |
| 3) | DFS, SCE | 69.74 | 61.02 | 71.09 | 67.28 | 64.68 | 76.93 | 70.81 |
| 4) | DFS, SDI | 70.68 | 61.55 | 71.36 | 67.86 | 65.38 | 77.16 | 71.27 |
| 5) | DFS, SCE, SDI | **71.24** | **62.26** | **71.91** | **68.74** | **66.03** | **77.90** | **71.97** |

Table 6: Impact of time step $T$.

| $T$ | G→C | G→B | G→G | Avg. |
|---|---|---|---|---|
| 2 | 68.97 | 59.83 | 70.35 | 66.38 |
| 3 | 70.15 | 60.24 | 70.85 | 67.08 |
| 4 | 70.82 | 60.77 | 71.06 | 67.55 |
| 5 | **71.24** | **62.26** | **71.91** | **68.74** |
| 6 | 71.03 | 61.90 | 71.38 | 68.10 |

**Results.** Table 2 reports the outcomes. The proposed method outperforms all the compared methods. Specifically, with the same EVA02-L VLM backbone, it outperforms the second-best MGRNet [67] by 1.71%, 1.12% and 1.94% in mIoU on the C, B, and M unseen domains, respectively, when using G as the source domain. It outperforms the second-best MGRNet [67] by 1.33%, and 1.47% in mIoU on the B, and M unseen domains, respectively, when using C as the source domain.

**Generalization on Various Foundation Models.** We further test the generalization ability of the proposed method when using other foundation models, namely, DINOv2 [51] and CLIP [59]. Since the proposed method requires the class-wise text as input, we use CLIP text encoder under all the experiments. The experiments are conducted when using GTA as the source domain. Table 3 reports the outcomes. The proposed method shows a better generalization ability over these foundation models than the prior arts.

**Effectiveness over Static Alignment.** To validate the effectiveness of the stochastic evolution in the proposed method, we compare it with two static alignment methods, namely, direct cosine matching (DCM) and without time-indexed dynamics (w.o. TID). The experiments are conducted when using GTA as the source domain. The results in Table 4 show that the stochastic evolution clearly outperforms both static alignment methods, indicating its contribution to the overall performance.

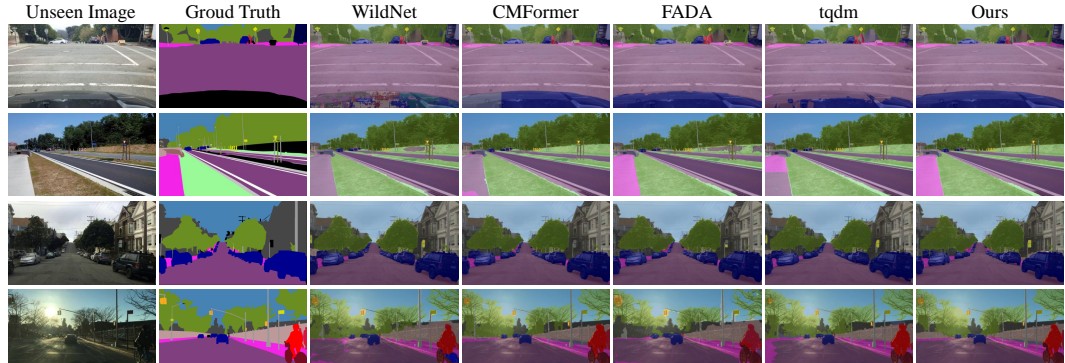

| Unseen Image | Groud Truth | WildNet | CMFormer | FADA | tqdm | Ours |

Figure 3: Exemplar segmentation results of existing DGSS methods (WildNet [38], CMFormer [9], FADA [8], tqdm [52]), and the proposed SBGen on unseen target domains.

Table 7: Impact of the optimal transport solving methods on generalization performance.

| Method | $G \rightarrow C$ | $G \rightarrow B$ | $G \rightarrow M$ | Avg. | $C \rightarrow B$ | $C \rightarrow M$ | Avg. |
|---|---|---|---|---|---|---|---|
| Baseline | 68.88 | 59.18 | 70.10 | 66.05 | 64.72 | 76.15 | 70.44 |
| CFM [69] | 70.73 | 61.05 | 70.64 | 67.47 | 64.81 | 76.57 | 70.69 |
| Sinkhorn | 69.62 | 60.18 | 70.57 | 66.79 | 65.29 | 76.48 | 70.89 |
| Ours | **71.24** | **62.26** | **71.91** | **68.74** | **66.03** | **77.90** | **71.97** |

Table 8: Computational cost analysis. The GPU hour refers to one single A100 GPU hardware.

| Method | GPU Hours | #Para. | Model Size |
|---|---|---|---|
| Baseline | 79.0 | 788.59M | 5.60GB |
| Ours | 79.2 | 790.17M | 5.61GB |

## 5.3 Ablation Studies

**On Each Component.** On top of a VFM and a task-specific head, the proposed method consists of three key components, namely, Domain-aware Visual Feature Selection (DFS), Stochastic Cross-Domain Evolution (SCE), and Stochastic Domain-Agnostic Interpolation (SDI). Table 5 leverages four experiment settings to inspect how each component impacts the overall performance. Overall, all the components contribute positively to the generalization performance. Specifically, DFS leads to an up to 0.47% mIoU improvement on GTA5→C/B/M (Avg.) setting. SCE further improves the performance by 0.76% mIoU on the same setting. SDI brings an additional 0.58% mIoU improvement, reaching the final performance of 68.74%.

**On Time Step $T$.** Table 6 further studies how the time step $T$ impacts the generalization performance. By default, $T$ is set to 5 under all of our experiments. We further test the situation when it is 2, 3, 4, 5, and 6. The results show that the generalization performance achieves the optimal when it is set to be 5. A too-small time step may lead to the under-training problem, while a too-large time step may already saturate the performance but lead to more computation overhead.

**Impact of Optimal Transport Solving.** We compare the proposed method with Conditional Flow Matching (CFM) [69] and the commonly-used Sinkhorn transport (Sinkhorn). Table 7 shows that these methods achieve a very similar result. The proposed method shows a slight improvement, which may be explained that it is more tailored for the alignment between image and text embeddings.

**Computational Cost Analysis.** We've compared the proposed method with the baseline in terms of the training time, parameter number and model size under the DGSS experimental setting. Table 8 shows that although the proposed method achieves an acceptable trade-off between computational cost and performance improvement over the baseline. Specifically, the increase of GPU hour is 0.2 hours, the parameter number increase is 1.58 million, and the model size increase is 0.01GB. The GPU hour refers to the A100 GPU hardware.

**On Hyper-parameter $\lambda$.** The hyper-parameter $\lambda$ in Eq.9 balances the impact of the task-specific loss and the cross-modal Schrödinger Bridge loss. To test its impact, we conduct the experiments when it is set 0.01, 0.1, 1, 10 and 100, respectively. The results in Table 9 show that when $\lambda$ is set to 1, the generalization performance achieves the optimal. A too small $\lambda$ (e.g., 0.01) may not impose a sufficient alignment between the domain-agnostic class text and the domain-specific visual features. A too large $\lambda$ (e.g., 100) may overwhelm the task loss, leading to a performance drop.

**On Feature Selection Ratio $K$.** By default, $K$ is set to be 0.3 under all of our experiments. To inspect its impact, we conduct the experiments when it is set to be 0, 0.1, 0.2, 0.4 and 0.5, respectively.

Table 9: Impact of hyper-parameter $\lambda$. Evaluation metric is mIoU in %.

| $\lambda$ | $G \to C$ | $G \to B$ | $G \to M$ | Avg. |
|---|---|---|---|---|
| 0.01 | 69.83 | 60.67 | 70.68 | 67.06 |
| 0.1 | 70.65 | 61.72 | 71.34 | 67.90 |
| 1 | **71.24** | **62.26** | **71.91** | **68.74** |
| 10 | 71.01 | 61.17 | 71.27 | 67.82 |
| 100 | 70.37 | 61.26 | 71.28 | 67.64 |

Table 10: Impact of hyper-parameter $K$. Evaluation metric is mIoU in %.

| $K$ | $G \to C$ | $G \to B$ | $G \to G$ | Avg. |
|---|---|---|---|---|
| 0 | 70.39 | 60.57 | 70.54 | 67.17 |
| 0.1 | 70.57 | 60.81 | 70.90 | 67.43 |
| 0.2 | 71.04 | 61.58 | 71.59 | 68.07 |
| 0.3 | **71.24** | **62.26** | **71.91** | **68.74** |
| 0.4 | 71.15 | 62.04 | 71.37 | 68.19 |
| 0.5 | 70.92 | 61.75 | 71.16 | 67.94 |

The results in Table 10 show that when $K$ is set to 0.3, the segmentation performance on unseen target domains achieves the optimal performance. A too small $K$ (*e.g.*, 0 and 0.1) may not select sufficient visual features to align with the domain-agnostic class features, which may under-fit the representation. A too large $K$ (*e.g.*, 0.4 and 0.5) may introduce more visual features that are not domain-specific, which results in a slight performance drop.

### 5.4 Qualitative Results

**On Visual Prediction Maps.** Fig. 3 displays some visual prediction maps on unseen CityScapes, BDD and MAP target domains, when using GTA5 as the source domain. The proposed method shows a more precise per-pixel prediction than existing state-of-the-art DGSS methods, namely, WildNet [38], CMFormer [9], FADA [8], and tqdm [52].

**On Feature Space.** We further inspect if the proposed method can alleviate the domain gap between the source domain and unseen target domains over the baseline. For each sample in each domain, we extract the features before the task-specific decoder, flatten them into a feature embedding, and then project the embedding into a latent space by t-SNE visualization. All the experiments are conducted under the $G \to$ C, B, M setting. As shown in Fig. 5, the samples from three unseen target domains are more uniformly distributed and aligned closer to the source domain by the proposed method, indicating its effectiveness to mitigate the domain gap by aligning the domain-specific visual features to the domain-agnostic text embedding.

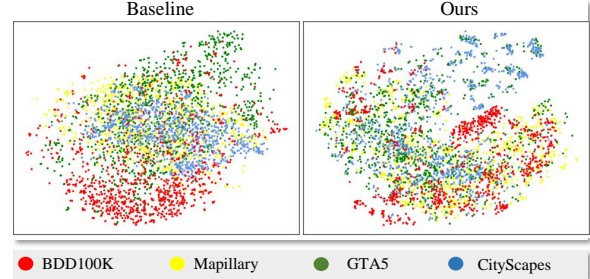

Figure 4: t-SNE visualization. Feature embedding is extracted from the last VFM layer. Left: baseline; Right: ours.

## 6 Conclusion

This paper introduced `SBGen`, a principled stochastic domain generalization framework, which bridges domain-specific image features and domain-agnostic textual semantics through Schrödinger Bridge dynamics. It leverages textual queries to guide visual feature selection and employs a time-conditioned stochastic evolution to model a continuous trajectory from source domain representations to semantic targets, enabling robust generalization to unseen target domain samples. Extensive experiments show its superiority over domain generalization in both classification and segmentation.

**Future Work, Limitation & Societal Impact.** Future work may explore extensions to multimodal generalization across more complex modalities (*e.g.*, audio and video), and efficient approximations of high-dimensional Schrödinger Bridge dynamics. However, the proposed SBGen requires a simulation of multiple-step stochastic differential equation (SDE) for each batch, which additionally adds multiple forward passes and increases the complexity over the baseline. Still, it exhibits a good trade-off between the complexity and the clear performance improvement on unseen target domains. This work can benefit domain generalization in various real-world applications, contributing to more reliable artificial intelligence systems. We do not envision its negative societal impact.

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

**Technical Appendices and Supplementary Material**

## Contents

# A  Theoretical Analysis: Generalization Error Bound

In this section, we derive a generalization error bound on the unseen target domains of the proposed `SBGen`, and demonstrate its superiority over the generalization error bound over the VLM baseline.

We start from some key definitions. Let $P_0$ and $P_1$ denote the source and the target feature distributions in $\mathbb{R}^C$. Let $\mathbb{Q}$ denote the law of our learned stochastic evolution (Schrödinger Bridge) from $P_0$ to $P_1$. The risk of a classification or segmentation model $h$ w.r.t. distribution $P$ can be defined as

$$R_P(h) \; = \; \mathbb{E}_{z \sim P}\big[\ell\big(h(z), y\big)\big], \tag{10}$$

where the task loss function $\ell$ is bounded in $[0, 1]$, and $y$ denotes the ground truth.

The analysis will be based on the deduction of the empirical error on source domain and the expected error on target domain, defined as

$$R_{P_1}(h) \quad \text{(target risk)} \quad \text{to} \quad R_{P_0}(h) \quad \text{(source risk)}. \tag{11}$$

**Lemma 1. Ben-David Transfer Bound.** *Let $P_0$ and $P_1$ be two distributions over a common feature space $\mathcal{Z} \subseteq \mathbb{R}^C$, corresponding to the source and target domains, respectively. Let $h : \mathcal{Z} \to \mathcal{Y}$ be a hypothesis, and let $\ell : \mathcal{Y} \times \mathcal{Y} \to [0, 1]$ be a bounded loss function. Then, the target risk of $h$ satisfies:*

$$R_{P_1}(h) \; \leq \; R_{P_0}(h) \; + \; \text{Distance}_{\text{TV}}(P_0, P_1) \; + \; \epsilon_{\text{joint}}, \tag{12}$$

*where $R_P(h) \; := \; \mathbb{E}_{(z,y) \sim P}[\ell(h(z), y)]$ denotes the expected risk under distribution $P$, $\text{Distance}_{\text{TV}}(P_0, P_1) := \frac{1}{2} \int |dP_0 - dP_1|$ denotes the total variation distance between the distributions, and $\epsilon_{\text{joint}} := \min_{h' \in \mathcal{H}} [R_{P_0}(h') + R_{P_1}(h')]$ denotes the joint risk of the optimal shared hypothesis.*

**Proof.** Please refer to [6] for the detailed proof.

**Theorem 1. Variation Distance via Schrödinger Bridge.** *Let $\mathbb{Q}$ be the solution to the Schrödinger Bridge problem between distributions $P_0$ and $P_1$ over $\mathbb{R}^C$, i.e., a path measure such that $\mathbb{Q}_{t=0} = P_0$ and $\mathbb{Q}_{t=1} = P_1$. Let $\mathbb{P}$ denote the reference Brownian motion with the same marginals at $t = 0$ and $t = 1$. Then the total variation distance between $P_0$ and $P_1$ is bounded by the KL divergence between $\mathbb{Q}$ and $\mathbb{P}$ as:*

$$\text{Distance}_{\text{TV}}(P_0, P_1) = \text{Distance}_{\text{TV}}(\mathbb{Q}_0, \mathbb{Q}_1) \leq \sqrt{\tfrac{1}{2}\text{KL}(\mathbb{Q} \, \| \, \mathbb{P})}. \tag{13}$$

**Proof:** We apply Pinsker's inequality to the marginals of the SB process:

$$\text{Distance}_{\text{TV}}(\mu, \nu) \leq \sqrt{\tfrac{1}{2}\text{KL}(\mu \, \| \, \nu)} \quad \text{for all probability measures } \mu, \nu. \tag{14}$$

Since the Schrödinger Bridge process $\mathbb{Q}$ interpolates from $P_0$ to $P_1$ over time $t \in [0, 1]$, and $\mathbb{Q}_0 = P_0$, $\mathbb{Q}_1 = P_1$, we apply Pinsker's inequality to the terminal marginal distributions of $\mathbb{Q}$ and $\mathbb{P}$.

Because $\mathbb{Q}$ and $\mathbb{P}$ are path measures with the same support, we have:

$$\text{Distance}_{\text{TV}}(\mathbb{Q}_0, \mathbb{Q}_1) \leq \sqrt{\tfrac{1}{2}\text{KL}(\mathbb{Q} \, \| \, \mathbb{P})}, \tag{15}$$

and by definition $\mathbb{Q}_0 = P_0$, $\mathbb{Q}_1 = P_1$, so:

$$\text{Distance}_{\text{TV}}(P_0, P_1) \leq \sqrt{\tfrac{1}{2}\text{KL}(\mathbb{Q} \, \| \, \mathbb{P})}. \tag{16}$$

**Theorem 2. Generalization Error Bound of Schrödinger Bridge.** *Let $P_0$ and $P_1$ be the source and target feature distributions over $\mathbb{R}^C$. Let $\mathbb{Q}_\theta$ be the path distribution induced by the Schrödinger Bridge model trained to transport $z_0 \sim P_0$ to $z_T \sim P_1$, and let $\mathbb{P}$ be the Brownian reference process. Let $h_\theta$ be the hypothesis (e.g., classifier or segmenter) composed with the SB mapping. Then, the expected target-domain risk is bounded as:*

$$R_{P_1}(h_\theta) \; \leq \; R_{P_0}(h_\theta) \; + \; \sqrt{\tfrac{1}{2}\text{KL}(\mathbb{Q}_\theta \, \| \, \mathbb{P})} \; + \; \epsilon_{\text{joint}}, \tag{17}$$

where $R_P(h) := \mathbb{E}_{(z,y)\sim P}[\ell(h(z), y)]$ is the expected risk under distribution $P$, and $\epsilon_{\text{joint}} := \min_{h'\in\mathcal{H}}[R_{P_0}(h') + R_{P_1}(h')]$ is the optimal joint risk over the hypothesis class.

**Proof:** From Lemma 1, the basic transfer bound gives:

$$R_{P_1}(h_\theta) \leq R_{P_0}(h_\theta) + \text{Distance}_{\text{TV}}(P_0, P_1) + \epsilon_{\text{joint}}. \tag{18}$$

From Theorem 1, we apply Pinsker's inequality to the SB marginals:

$$\text{Distance}_{\text{TV}}(P_0, P_1) \leq \sqrt{\tfrac{1}{2}\text{KL}(\mathbb{Q}_\theta\|\mathbb{P})}. \tag{19}$$

Substituting yields:

$$R_{P_1}(h_\theta) \leq R_{P_0}(h_\theta) + \sqrt{\tfrac{1}{2}\text{KL}(\mathbb{Q}_\theta\|\mathbb{P})} + \epsilon_{\text{joint}}. \tag{20}$$

**Theorem 3. Tighter Generalization Bound for Schrödinger Bridge Model.** *Let $P_0$ and $P_1$ be the source and target distributions over $\mathbb{R}^C$. Let $\mathbb{Q}_\theta$ denote the Schrödinger Bridge process that evolves samples from $P_0$ to $P_1$ with reference prior $\mathbb{P}$. Let $M_\phi : \mathbb{R}^C \to \mathbb{R}^C$ be a deterministic baseline transport (e.g., cosine projection or prompt-aligned mapping), and let $P_1^\phi := M_{\phi\#}P_0$ denote the induced pushforward distribution. Let $\ell$ be a bounded loss function and $h_\theta$, $h_\phi$ the hypotheses composed with the SB and baseline mappings, respectively. Then the generalization error of the SB model satisfies a strictly tighter upper bound:*

$$R_{P_1}(h_\theta) \leq R_{P_0}(h_\theta) + \sqrt{\tfrac{1}{2}\text{KL}(\mathbb{Q}_\theta\|\mathbb{P})} + \epsilon_{\text{joint}}, \tag{21}$$

$$R_{P_1}(h_\phi) \leq R_{P_0}(h_\phi) + \text{Distance}_{\text{TV}}(P_0, P_1^\phi) + \epsilon_{\text{joint}}. \tag{22}$$

*Moreover, since $\mathbb{Q}_\theta$ minimizes the entropy-regularized transport cost from $P_0$ to $P_1$, and $M_\phi$ induces a deterministic coupling,*

$$\sqrt{\tfrac{1}{2}\text{KL}(\mathbb{Q}_\theta\|\mathbb{P})} < \text{Distance}_{\text{TV}}(P_0, P_1^\phi) \tag{23}$$

*unless $M_\phi$ itself induces the SB-optimal coupling.*

**Proof:** The bound for the SB model is established in Theorem 2. For the deterministic baseline, we consider the mapping $z_1 = M_\phi(z_0)$ and define $P_1^\phi := M_{\phi\#}P_0$ as the transformed distribution.

Using the basic transfer bound (Lemma 1) again:

$$R_{P_1}(h_\phi) \leq R_{P_0}(h_\phi) + \text{Distance}_{\text{TV}}(P_0, P_1^\phi) + \epsilon_{\text{joint}}. \tag{24}$$

In contrast, the SB model produces a path distribution $\mathbb{Q}_\theta$ over $z_t$ such that $\mathbb{Q}_{t=0} = P_0$, $\mathbb{Q}_{t=1} = P_1$. Applying Pinsker's inequality as in Theorem 2, we have:

$$\text{Distance}_{\text{TV}}(P_0, P_1) \leq \sqrt{\tfrac{1}{2}\text{KL}(\mathbb{Q}_\theta\|\mathbb{P})}. \tag{25}$$

Since the Schrödinger Bridge is known to minimize the KL divergence over all couplings between $P_0$ and $P_1$, and the deterministic map $M_\phi$ induces a coupling $\pi^\phi(z_0, z_1) = \delta(z_1 - M_\phi(z_0))$, we have:

$$\text{KL}(\mathbb{Q}_\theta\|\mathbb{P}) < \text{KL}(\pi^\phi\|\mathcal{R}), \tag{26}$$

for any reference coupling $\mathcal{R}$, unless $\pi^\phi$ itself is the SB-optimal coupling.

Therefore, the divergence and the TV-based generalization bound is strictly tighter under the SB transport.

**Corollary 1. Match of the generalization bound between the SB model and the Deterministic Baseline.** *Under the assumptions of Theorem 3, the generalization bounds of the Schrödinger Bridge model and the deterministic baseline coincide if and only if the SB-induced coupling $\mathbb{Q}_\theta$ corresponds to a deterministic map $M^*$ satisfying:*

$$\mathbb{Q}_\theta(z_0, z_1) = \delta(z_1 - M^*(z_0)) \cdot P_0(z_0), \tag{27}$$

*and this map $M^*$ pushes $P_0$ exactly onto $P_1$, i.e.,*

$$M_\#^* P_0 = P_1. \tag{28}$$

---

**Algorithm 1** Schrödinger Bridge-Guided Domain Generalization

---

**Require:** Source images $\{x_i\}_{i=1}^N$, class text queries $\{\mathcal{Q}_c\}_{c=1}^C$, vision encoder $\mathcal{E}$, text encoder $\mathcal{T}$, time horizon $T$, noise scale $\varepsilon$, number of steps $L$
**Ensure:** Learned drift model $\mathcal{U}_\theta$, prediction decoder $\mathcal{D}$

1: Initialize $\mathcal{U}_\theta$, $\mathcal{D}$
2: **for** each training iteration **do**
3:     Sample mini-batch $\{x_i, y_i\}_{i=1}^B$ from source domain
4:     ### Domain-aware Visual Feature Selection ###
5:     Extract dense visual features: $\mathcal{F}_i = \mathcal{E}(x_i)$
6:     Encode class queries: $q_c = \mathcal{T}(\mathcal{Q}_c)$
7:     Compute similarity scores $S_{h,w,c} = \langle \mathcal{F}_{h,w}, q_c \rangle$
8:     Select top-$k$ features: $\mathcal{F}_s \leftarrow$ query-guided selection from $\mathcal{F}$
9:     **for** each feature vector $z_0 \in \mathcal{F}_s$ **do**
10:         Initialize $z_t \leftarrow z_0$
11:         **for** $l = 1$ to $L$ **do**
12:             $t \leftarrow \frac{l}{L}$
13:             Sample noise $\xi \sim \mathcal{N}(0, I)$
14:             ### Stochastic Cross-Domain Evolution & Domain-Agnostic Interpolation ###
15:             Update: $z_t \leftarrow z_t + \mathcal{U}_\theta(z_t, t)\, \Delta t + \sqrt{2\varepsilon\, \Delta t}\, \xi$
16:         **end for**
17:         Store final evolved feature $z_T$
18:     **end for**
19:     ### Prediction Head ###
20:     Predict: $\hat{y}_{\text{cls}}, \hat{y}_{\text{seg}} \leftarrow \mathcal{D}(\{z_T\}, \{q_c\})$
21:     Compute task losses $\mathcal{L}_{\text{sup}}$
22:     Estimate SB divergence (e.g., via score matching or IPFP): $\mathcal{L}_{\text{SB}}$
23:     Update parameters via $\nabla_\theta(\mathcal{L}_{\text{sup}} + \lambda \cdot \text{KL}(\mathbb{Q}_\theta | \mathbb{P}))$
24: **end for**

---

*In this case, the KL divergence collapses to:*

$$\text{KL}(\mathbb{Q}_\theta \| \mathbb{P}) = 2 \cdot \text{Distance}_{\text{TV}}{}^2(P_0, P_1), \tag{29}$$

*and the generalization bounds for both models are equal:*

$$R_{P_1}(h_\theta) = R_{P_1}(h_\phi). \tag{30}$$

We conclude this section by the following remark. The proposed SBGen, a Schrödinger Bridge guided framework, not only provides a principled dynamic interpolation between source and target distributions but also holds a strictly tighter generalization error upper bound compared to the deterministic baseline.

## B   Pseudo-code: Schrödinger Bridge-Guided Domain Generalization

A pseudo-code implementation of the proposed SBGen is given in Algorithm 1.

## C   More Implementation Details

Following prior work [52], the same training configuration is set for all types of pre-trained foundation models (*e.g.*, CLIP, DINOv2, and EVA02), and for both domain generalization in classification and semantic segmentation.

In all the experiments, the images are cropped and resized into 512×512 pixels. The batch size is set 16, with an AdamW optimizer. The initial learning rate is set to be $1 \times 10^{-5}$ for all the synthetic-to-real settings, and is set to be $1 \times 10^{-4}$ for all the real-to-real settings. The learning rate of the backbone is further scaled by 0.1. The training does not terminate after 20,000 iterations. Following [52], a linear warm-up is applied after 1500 iterations, followed by a linear decay. Some common data augmentation techniques, namely, random scaling, random cropping, random flipping, color jittering, and rare class sampling, are also used.

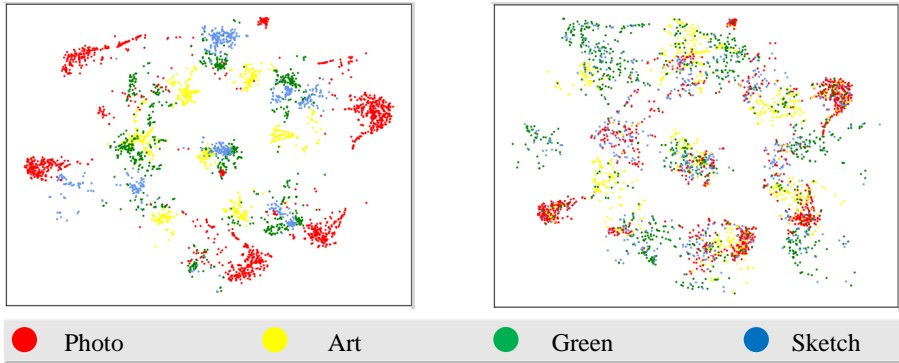

| ● Photo | ● Art | ● Green | ● Sketch |

Figure 5: t-SNE visualization. Feature embedding is extracted before the decoder. Left: EVA02 baseline; Right: ours.

**Domain generalization in classification.** For the classification task, the image encoder $\mathcal{E}$ and the text encoder $\mathcal{T}$ use the pre-trained CLIP in align with the prior DG methods. The task-specific decoder $\mathcal{D}$ is a linear layer followed by a Softmax layer.

**Domain generalization in segmentation.** Following prior domain generalized semantic segmentation methods [52, 67], the default image encoder $\mathcal{E}$ and the text encoder $\mathcal{T}$ use the pre-trained EVA-02 [25]. The image encoder $\mathcal{E}$ can also be switched to CLIP, SAM and DINOv2 in our experiments. The task-specific decoder $\mathcal{D}$ integrates the pixel decoder of the Mask2Former model [14].

## D  More Feature Space Analysis

Fig.4 in the main text inspects whether the proposed SBGen can improve the generalization ability over the baseline, on the task of domain generalized semantic segmentation (DGSS). In the supplementary material, we further inspect whether the proposed SBGen can improve the generalization ability over the baseline on domain generalization in the classification task.

Specifically, we extract the feature of each sample from the PACS dataset before the decoder and concatenate it into a feature vector. Then, we display the feature vector of each sample regardless of the domain identity by t-SNE visualization. Feature vectors from the Photo, Art Painting, Cartoon and Sketch domains are colored in red, yellow, green and blue, respectively.

The feature space of the original baseline and the proposed SBGen is visualized in the left and right of Fig. 5, respectively. In each cluster that shares the same semantic category, the samples from different domains are more uniformly distributed by the proposed SBGen, indicating its effectiveness to mitigate the domain gap.

## E  More Visual Prediction Results

Fig. 6 shows more results under G → B, M, C setting. The segmentation results show that the proposed SBGen shows better pixel-wise prediction than the compared DGSS methods, especially in terms of the completeness of objects.

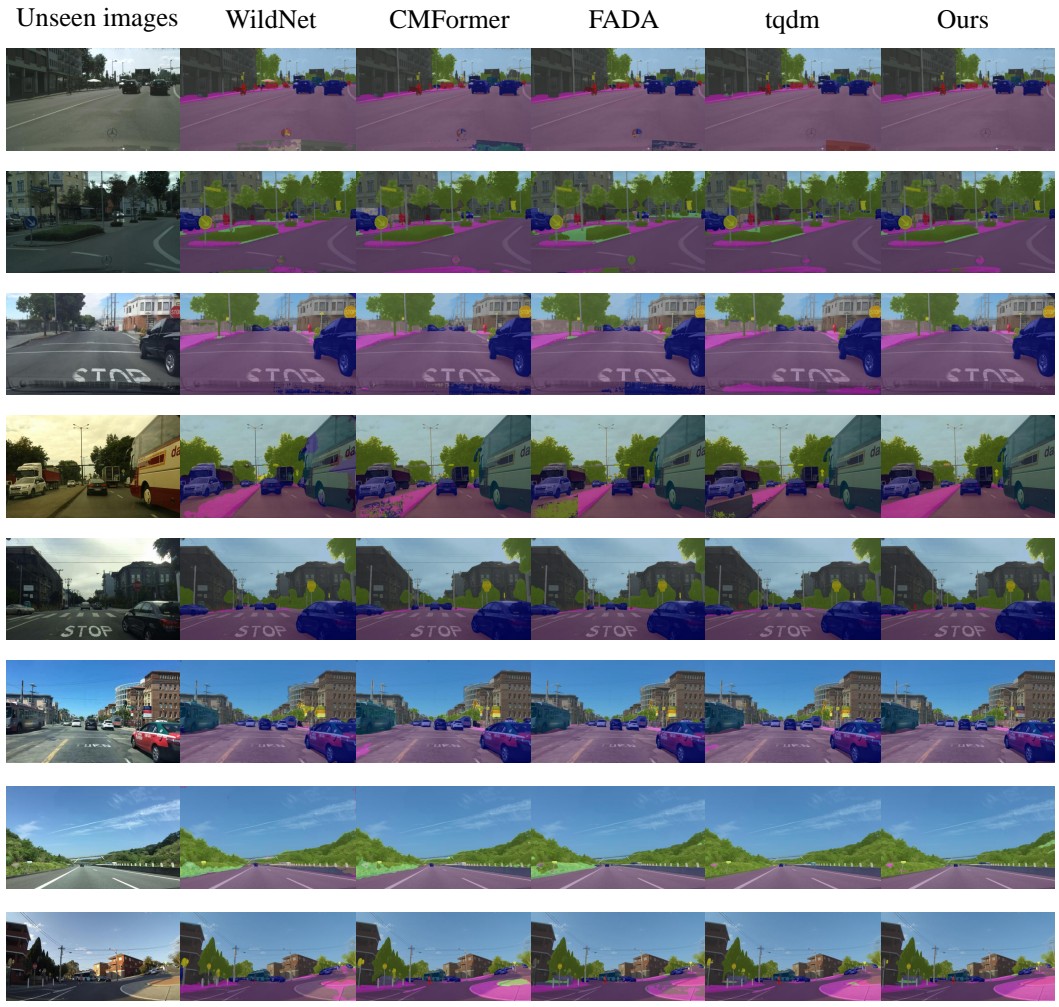

| Unseen images | WildNet | CMFormer | FADA | tqdm | Ours |

Figure 6: Visual segmentation results on unseen target domains under the G → B, M, C setting. The proposed SBGen is compared with WildNet [38], CMFormer [9], FADA [8], and tqdm [52].

