# OpenReview forum: "Learning a Cross-Modal Schrödinger Bridge for Visual Domain Generalization"
_NeurIPS.cc/2025/Conference — NeurIPS 2025 poster_

### Official Review · Reviewer_3EAi · 2025-06-04

**Clarity:** 3
**Significance:** 3
**Originality:** 4
**Rating:** 5
**Confidence:** 3

**Summary:**

This paper addresses the limitations of static alignment strategies in vision-language domain generalization by proposing a novel framework, SBGen, which models the alignment between domain-specific visual features and domain-agnostic textual semantics as a stochastic evolution process via a Schrödinger Bridge formulation. The method leverages a time-indexed stochastic differential equation to progressively transform visual features toward semantic anchors. The proposed framework is evaluated on standard domain generalization benchmarks for both classification and segmentation tasks and achieves consistent improvements over strong baselines. While the approach is theoretically well-grounded and empirically effective, the experimental validation could be further strengthened by isolating the contribution of the stochastic evolution process.

**Questions:**

1. Can the authors include an ablation study that replaces the stochastic evolution with a static alignment baseline, such as direct cosine matching or prompt learning without time-indexed dynamics? This would clarify the contribution of the stochastic modeling to the overall performance.
2. Could the authors provide an analysis of the stochastic evolution paths, such as visualizations of intermediate feature distributions or quantitative metrics like Wasserstein distance over time? This would help assess whether the feature trajectories are smooth, stable, and semantically meaningful.
3. Given that the training process involves stochastic modeling, is it possible to extend the inference stage to quantify uncertainty, for example through sampling multiple evolution paths? Such an addition could enhance the robustness of predictions under domain shift.

**Ethical Concerns:**

["NO or VERY MINOR ethics concerns only"]

**Final Justification:**

The authors provided clear, direct, and convincing responses. Based on their effective rebuttal and my continued recognition of the paper's core strengths, I will maintain my initial positive assessment and rating.

**Limitations:**

yes

**Paper Formatting Concerns:**

The tables in the paper include vertical rules, which do not comply with the publication formatting guidelines of NeurIPS.

**Quality:**

3

**Strengths And Weaknesses:**

The motivation of the paper is clear and well-justified. The authors identify a key limitation in existing vision-language DG methods—the reliance on static alignment—which may fail under severe distribution shifts. In response, the paper proposes modeling the alignment as a stochastic semantic evolution, providing a more flexible and theoretically grounded solution. The experimental evaluation is extensive, covering a wide range of datasets and tasks. The method achieves strong empirical results, outperforming state-of-the-art baselines in both image classification and semantic segmentation. The paper also includes ablations on hyperparameters and feature selection, adding robustness to the empirical claims. However, a significant limitation lies in the lack of direct ablation isolating the impact of the proposed stochastic evolution process. While the ablations adjust loss balancing factors and feature selection, they do not explicitly compare the stochastic evolution against a purely static alignment baseline. This omission makes it difficult to rigorously assess how much of the observed improvement is due to the dynamic modeling itself. Additionally, the stochastic process is not analyzed in terms of stability, convergence, or intermediate behavior, and inference remains deterministic without leveraging uncertainty modeling, leaving some aspects of the framework under-explored.

---

> ### Author Rebuttal · Authors · 2025-07-30
>
> We appreciate the reviewer for the positive comments on the clear motivation, technical novelties, extensive experiments and strong performance. The below is a point-by-point response to provide more ablation studies, more detailed analysis on stochastic evolution paths and uncertainty modeling.
>
> **Q1**: More ablation studies on static alignment baselines, such as direct cosine matching or prompt learning without time-indexed dynamics.
>
> **R**: Per your suggestion, we've added the experiments when replacing the stochastic evolution with two static alignment methods, namely, direct cosine matching (DCM) and without time-indexed dynamics (w.o. TID).
> The experiments are conducted when using GTA as the source domain.
> The results in the attached table~show the stochastic evolution clearly outperforms both two methods, indicating its contribution to the overall performance.
>
> | Method | G->C | G->B | G->M | Avg. |
> | ------- | ------ | ----- | ------- | ------ |
> | Baseline | 68.88 | 59.18 | 70.10 | 66.05 |
> | DCM | 69.78 | 60.92 | 70.84 | 67.18 |
> | w.o. TID | 70.01 | 61.16 | 71.13 | 67.43 |
> | Ours | **71.24** | **62.26** | **71.91** | **68.74** |
>
> **Q2**: More analysis of the stochastic evolution paths, such as visualizations of intermediate feature distributions or quantitative metrics like Wasserstein distance over time?
>
> **R**: Many thanks for your valuable suggestion.
> Unfortunately, this year NeurIPS does not allow the submission of visualization in 1-page PDF or external links.
> After consulting the program chair, we are allowed to describe the visualization method and the outcome in text.
> We extract the intermediate feature embeddings from both the text embedding and image embedding over different time steps during the stochastic evolution.
> We then display them in the feature space by t-SNE visualization.
> It is observed that the intermediate features from the text embedding and the image embedding are gradually more uniformly mixed, indicating it is stable and domain meaningful.
> We also compute the Wasserstein distance over time. The results attached below~show that the Wasserstein distance metric is becoming smaller over time, indicating the feature trajectories are converging, smooth and stable.
> Both results are promised to be added in the revised manuscript.
>
> | Time $T$ | 1 | 2 | 3 | 4 | 5 |
> | ------- | ------ | ----- | ------- | ------ | ------ |
> | Distance | 1.3367 | 1.3046 | 1.2751 | 1.2409 | 1.2210 |
>
> **Q3**: Is it possible to extend the inference stage to quantify uncertainty, for example through sampling multiple evolution paths? Such an addition could enhance the robustness of predictions under domain shift.
>
> **R**: To extend the robustness of inference under domain shift, we leverage the inherent stochasticity of the Schrödinger Bridge evolution to model uncertainty in the prediction stage. Specifically, we simulate multiple independent realizations of the stochastic process, forming a Monte Carlo ensemble of evolved features.
>
>
>
> **Monte Carlo Evolution Paths**.  Given the initial visual tokens  $\{\boldsymbol{z} _{0}\} \sim P _{0}$, we simulate $M$ independent evolution paths by sampling distinct noise trajectories $\{\xi _{n}^{(m)}\} _{n=0}^{T-1}$, where for each path we have $m = 1, \ldots, M$. The terminal state of each path is computed via:
>
> $$
> \boldsymbol{z} _{t _T} ^{(m)} = \boldsymbol{z} _{t _0} + \sum _{n=0}^{T-1} f _\theta(z _{t _n}^{(m)}, t _n) \Delta t + \sum _{n=0}^{T-1} \sqrt{2\varepsilon \Delta t} \xi _n^{(m)},
> $$
> where each $\boldsymbol{z} _{t_n}^{(m)}$ is recursively computed from its own noise realization.
>
>
> **Prediction Mean and Uncertainty**.  We perform prediction using the decoder $\mathcal{D}$ over each sample path:
>
> $$ \hat{y}^{(m)} = \mathcal{D}(\mathcal{F}'^{(m)}, \{\boldsymbol{q}_{c}\}), $$
>
> and estimate the predictive mean and variance:
>
> $$ \bar{y} = \frac{1}{M} \sum_{m=1}^{M} \hat{y}^{(m)}, $$
>
> $$ \mathbb{V}(y) = \frac{1}{M} \sum_{m=1}^M (\hat{y}^{(m)} - \bar{y})^2.$$
>
> The variance $\mathbb{V}(y)$ quantifies the epistemic uncertainty, providing a confidence-aware prediction strategy across domains.
>
> As a result, the variance $\mathbb{V}(y)$ is used to report the uncertainty score.
> The lower variance refers to more confident predictions and less uncertainty across domains.
> We report the uncertainty score on the domain generalized semantic segmentation, when we conduct 5, 10, 15, 20, 25 and 30 Monte Carlo runs.
> The attached table~shows that the proposed method does not show clearly higher variance over the baseline, indicating it is robust to domain shift.
>
> | Run Number | 5 | 10 | 15 | 20 | 25 | 30 |
> | ------- | ------ | ----- | ------- | ------ | ------ |  ------ |
> | Baseline | 4.9439e-05 | 3.7132e-05 | 2.8410e-05 | 1.5496e-05 | 8.1127e-06 | 7.6321e-06 |
> | Ours (full) | 4.0271e-05 | 3.5851e-05 | 2.3389e-05 | 1.1906e-05 | 7.1032e-06 | 6.7117e-06 |
>
> Should you have further comments or questions, we are happy to address during the author-reviewer discussion stage.

---

> > ### Comment · Reviewer_3EAi · 2025-08-04
> >
> > The authors have provided clear and satisfactory responses to the concerns raised. Their clarifications further reinforce the strengths of the work. I maintain my original positive score.

---

> > > ### Author Response · Authors · 2025-08-04
> > > **Re: Official Comment by Reviewer 3EAi**
> > >
> > > Thanks for your positive recommendation. We will incooperate all the suggestions when revising our manuscript.

---

### Official Review · Reviewer_WcC6 · 2025-06-10

**Clarity:** 1
**Significance:** 2
**Originality:** 2
**Rating:** 2
**Confidence:** 4

**Summary:**

This paper proposes a method for domain generalization by learning a Schrödinger Bridge between visual and textual embeddings. The approach is designed to work for both classification and segmentation tasks. The authors also provide a theoretical analysis of the generalization error bound, supported by supplementary materials.

**Questions:**

See Weaknesses.

**Ethical Concerns:**

["NO or VERY MINOR ethics concerns only"]

**Final Justification:**

As I have addressed, the current version of this paper may need major revisions regarding those flaws and require another round of review. I will retain my rating.

**Limitations:**

See Weaknesses.

**Paper Formatting Concerns:**

No concerns regarding formatting issues.

**Quality:**

2

**Strengths And Weaknesses:**

*Strengths*

- The paper presents extensive experimental results across multiple tasks.

-  I have reviewed the supplementary materials and find that the main theoretical analysis regarding the generalization error bound appears to be technically sound.

*Weaknesses*

Several serious concerns remain about the problem setting and experimental details, which require further clarification.

- The paper seems to conflate the concepts of robustness and generalization, which are related but distinct; the authors are encouraged to use these terms more precisely and consistently throughout the paper.

- As shown in Figure 2, the target domain images seem to be involved in the Schrödinger Bridge learning process. **If this figure accurately reflects the methodology, it contradicts the standard domain generalization setting where target domain data should not be accessible during training.** This potential data leakage casts serious doubt on the reported improvements. If, on the other hand, the figure is inaccurate, it reflects a lack of clarity and attention to detail, suggesting the paper requires major revision to reach publication quality.

- The notational system is quite disorganized. For instance, the symbol $T$ is used to denote at least three different concepts: the deterministic transport map (e.g., Line 108), the target domain in $P^T$, and the number of steps or time horizon (e.g., Line 186). Furthermore, Figure 2 uses discrete time indices “$t = 1, 2, ..., T$”, whereas the rest of the paper repeatedly describes $t \in [0, 1]$ as a continuous interval. This inconsistency is confusing and undermines the clarity of the presentation. A thorough proofreading and cleanup of the notation is strongly recommended.

- In the checklist, the authors claim that limitations are discussed in the conclusion. However, I could not find any explicit discussion of the method’s limitations in that section.

- Around line 208, the KL loss is computed as an accumulation from step 0 to $T-1$. This iterative formulation likely incurs significant computational overhead. However, the paper does not provide any analysis or discussion regarding its computational cost or efficiency. Including such analysis would strengthen the work.

- The proposed method involves stochastic processes and may be sensitive to random seed choices. Reporting results averaged over multiple seeds or showing variance across runs would help assess the method's stability.

- Around line 291, the authors state that “as shown in Fig. 4, the samples from three unseen target domains are more uniformly distributed and aligned closer to the source domain by the proposed method.” However, from my perspective, the visualization alone does not convincingly support this claim. This might be due to the fact that segmentation datasets often contain diverse object types within a single image, making the domain gap difficult to visualize meaningfully. It is possible that visualizations based on embeddings from classification tasks might better illustrate the claimed alignment effect.

- In Supplementary Algorithm 1, the loss term $\mathcal{L}_{SB}$ is used but never explicitly defined.



Overall, though theoretical analysis is provided, the paper lacks careful writing, and significant revisions are needed. Therefore, I recommend rejection at this stage.

---

> ### Author Rebuttal · Authors · 2025-07-30
>
> We appreciate the reviewer for the positive comments on the theoretical analysis and extensive experiments. The below is a point-by-point response to clarify the notations, experimental settings and other presentation issues.
>
> **Q1**: Confusion between *robustness* and *generalization*.
>
> **R**: We are sorry for the confusion. We will replace the use of *robustness* to *generalization* throughout the paper for consistency.
>
> **Q2**: Clarity of Fig.2. Does it follow the domain generalization assumption?
>
> **R**： We are terribly sorry for the presentation of Fig.2, which is less clear than intended to properly demonstrate the domain generalization setting.
> To clarify, as detailed in the experimental section, the data from the target domain(s) is only involved in the inference stage for testing, not fed into the model in the training stage.It follows the basic assumption of the domain generalization. We will use:
>
> - put the data from the source/target domain into a box of solid/dashed line, respectively;
>
> - use the solid/dashed arrows of the data propogation to represent the input in the training/inference stage, respectively, where only source/target domain is involved.
>
> **Q3**： Notation system. $T$ has been defined three times.
>
> **R**: Many thanks for your careful checking, and we are sorry for the confusion in the notation system.
> For clarity and for correctness:
>
> - We will only use $T$ to denote the time indices.
>
> - Instead, we will use $\mathcal{M}$ to denote the deterministic transport map to distinguish.
>
> - We will replace $P^T$ by $P^U$ to denote the distribution of the unseen target domain.
>
> - We will remove the continuous interval ($t \in [0,1]$) and only keep the discrete interval for consistency.
>
> **Q4**: Missing explicit limitation discussion.
>
> **R**: Thanks for your careful checking and we are sorry for the ignorance. We will add the following text to explicitly discuss the limitation of the proposed method, written as:
>
> *However, the proposed SBGen requires a simulation of multiple-step stochastic differential equation (SDE) for each batch, which additionally adds multiple forward passes and increases the complexity over the baseline. Still, it exhibits a good trade-off between the complexity and the clear performance improvement on unseen target domains.*
>
> **Q5**: Computational cost analysis.
>
> **R**: Per your suggestion, we've accordingly compared the proposed method with the baseline in terms of the training time, parameter number and model size under the DGSS experimental setting. The attached table~shows that although the proposed method achieves an acceptable trade-off between computational cost and performance improvement over the baseline. Specifically, the increase of GPU hour is 0.2 hours, the parameter number increase is 1.58 million, and the model size increase is 0.01GB. The GPU hour refers to the A100 GPU hardware.
>
> | Method | Training Time | Parameter Number | Model Size |  G->C | G->B | G->M | Avg. |
> | ------ | ----- | ------- | ------ | ----- | ------- | ------ | ----- |
> | Baseline | 79.0 GPU hours | 788.59M | 5.60GB | 68.88 | 59.18 | 70.10 | 66.05 |
> | Ours | 79.2 GPU hours | 790.17M | 5.61GB | **71.24** | **62.26** | **71.91** | **68.74** |
>
> **Q6**: Stability, ideally justified by the variance of multiple runs.
>
> **R**: Many thanks for your valuable suggestion.
> We would like to kindly remind the reviewer that, the result is an average of three independent runs for domain generalized segmentation. We therefore report the standard deviation of these three runs from both the proposed method and the baseline.
> As shown below, the proposed method shows stability across independent runs.
>
> | Method | G->C | G->B | G->M | C->B | C->M |
> | ------ | ----- | ------- | ------ | ----- | ------- |
> | Baseline | 68.88$\pm$0.78 | 59.18$\pm$0.62 | 70.10$\pm$0.56 | 64.72$\pm$0.32 | 76.15$\pm$0.67 |
> | Ours | **71.24**$\pm$0.76 | **62.26**$\pm$0.45 | **71.91**$\pm$0.58 | **66.03**$\pm$0.42 | **77.90**$\pm$0.46 |
>
> **Q7**: t-SNE visualization in classification is more meaningful than segmentation.
>
> **R**: Thanks for your valuable suggestion. Unfortunately this year NeurIPS does not allow any PDF attachment or external link.
> After consulting the program chair, we describe the visualization method and the result.
> The classification experiments are conducted when PACS is used as the unseen target domain.
> The features from the last layer of the encoder are extracted and projected into the latent space for t-SNE visualization.
> It is observed that the per-class samples from our method are more uniformly distributed than the baseline.
> The updated figure will be incorporated into the revised version.
>
> Should you have further comments or questions, we are happy to address during the author-reviewer discussion stage.

---

> > ### Comment · Reviewer_WcC6 · 2025-08-01
> > **Reponse to rebuttal**
> >
> > Thank you for the rebuttal. Although the authors have acknowledged the flaws in both the mathematical formulation and presentation, these issues should ideally have been addressed prior to submission, and the current version does not meet the standards of the venue.
> > I believe very substantial revisions are necessary before the paper can be considered for acceptance. The revised version would also warrant another round of review. Therefore, I will keep my rating.

---

> ### Author Response · Authors · 2025-08-01
> **Re: Response to Reviewer WcC6**
>
> Thanks for your swift response. Should you find any other concerns that remain unaddressed, we are happy to address during the following week.

---

### Official Review · Reviewer_FGsB · 2025-07-02

**Clarity:** 3
**Significance:** 3
**Originality:** 3
**Rating:** 3
**Confidence:** 3

**Summary:**

This paper tackles the challenge of domain generalization (DG) in computer vision by leveraging vision-language models (VLMs) for improved robustness to unseen domains.  SBGen is introduced as a novel framework that formulates cross-modal alignment as a Schrödinger Bridge (SB) problem – essentially an entropy-regularized optimal transport between the source-domain image feature distribution and the target textual semantic distribution.  The approach consists of three key stages: (1) Text-guided domain-aware feature selection, which isolates the most semantically relevant image tokens per class by selecting top-$k$ spatial features that best match the class’s text embedding.  This focuses the model on domain-invariant content and removes background or domain-specific noise.  (2) Stochastic cross-domain evolution, which learns a time-dependent drift function to gradually transform the selected image features towards the text embedding distribution via a stochastic differential equation (SDE) – effectively simulating the Schrödinger Bridge dynamics between modalities.  (3) Stochastic domain-agnostic interpolation, which explicitly models and utilizes the continuous trajectory of intermediate features produced by the SDE, rather than only the end-points, to ensure a smooth semantic evolution and robust alignment.  After this evolution, the evolved features (now closer to domain-agnostic semantics) are injected back into the full image feature map and used for prediction by the task head.

**Questions:**

How did you choose the number of top-$k$ features for the domain-aware selection stage, and how sensitive is the performance to this choice?

**Ethical Concerns:**

["NO or VERY MINOR ethics concerns only"]

**Final Justification:**

I will maintain my score. My primary reason is that the contribution of this line of work to the community appears limited, particularly since many generalization issues can be addressed by scaling model size and training data. Moreover, it remains unclear whether the proposed method is applicable at larger scales.

**Limitations:**

yes

**Quality:**

3

**Strengths And Weaknesses:**

Strengths
The paper introduces a novel perspective on domain generalization by using the Schrödinger Bridge formulation to connect image and text feature distributions. This is a fresh idea that extends beyond conventional static alignment.

Weaknesses:

1. The SBGen pipeline adds non-trivial complexity to the training process. It requires simulating a multi-step SDE for each batch, which involves multiple forward passes through the drift network and adding noise.

2. Some design choices in SBGen are not exhaustively explored in the paper. For instance, the number of top-$k$ features selected per class (domain-aware selection) is presumably a hyperparameter – the method’s performance could depend on this, especially for segmentation where $k$ might decide how much of each object is covered by the stochastic alignment. The paper does not mention how $k$ was chosen or tuned. Am I missed?

3. The evaluation, while broad, still focuses on certain standard scenarios. For example, in segmentation, they tried single-source generalization (one source domain at a time). An interesting scenario could be multi-source DG for segmentation. SBGen in principle could handle multiple source domains by just pooling them into $P_S$, but whether that offers further gains or poses challenges (like needing to balance multiple domain biases) wasn’t explored.

4. The ablation studies provided are thorough for segmentation, but the paper does not explicitly show ablation results for the classification benchmarks. It would strengthen the paper to know if the three components (DFS, SCE, SDI) similarly contribute on classification accuracy.

---

> ### Author Rebuttal · Authors · 2025-07-30
>
> We appreciate the reviewer for acknowledging its novelties over existing methods to align the text and image features. The below is a point-by-point response to clarify the computational cost, hyper-parameter selection and to provide more experimental analysis.
>
> **Q1**: Computational Cost.
>
> **R**: Thanks for your insightful comments. We definitely agree with the reviewer that, as the proposed method simulates a multi-step SDE. To inspect its impact, we compute the training time, GFLOPs and model size between the proposed method and the baseline.
> The outcomes in the attached table show that, the proposed method only moderately increases the computation complexity but leads to a clear performance improvement, achieving a good trade-off between them.
> Specifically, the increase of GPU hour is 0.2 hours, the parameter number increase is 1.58 million, and the model size increase is 0.01GB. The GPU hour refers to the A100 GPU hardware.
>
> | Method | Training Time | Parameter Number | Model Size |  G->C | G->B | G->M | Avg. |
> | ------ | ----- | ------- | ------ | ----- | ------- | ------ | ----- |
> | Baseline | 79.0 GPU hours |	788.59M	| 5.60GB | 68.88 | 59.18 | 70.10 | 66.05 |
> | Ours | 79.2 GPU hours	| 790.17M | 5.61GB | **71.24** | **62.26** | **71.91** | **68.74** |
>
> **Q2**: Sensitivity of hyper-parameter $k$.
>
> **R**: We are sorry for the formatting of the supplementary material, which is less clear than intended.
> Specifically, we've conducted an experiment to test how the selection of $K$ impacts the performance of domain generalized semantic segmentation.
> The attached table shows that, a too small $K$ (\textit{e.g.}, 0.1) may not be able to select sufficient visual features that are domain-specific for the alignment between the domain-agnostic class text and the domain-specific visual features, which may under-fit the generalization representation.
> A too large $K$ (\textit{e.g.}, 0.4 and 0.5) may introduce more visual features that are not domain-specific in the representation learning, which may lead to over-fit and result in a slight performance drop.
>
> | $K$ | G->C | G->B | G->G | Avg. |
> | ------ | ----- | ------- | ------ | ----- |
> | 0.1 | 70.57 | 60.81 | 70.90 | 67.43 |
> | 0.2 | 71.04 | 61.58 | 71.59 | 68.07 |
> | 0.3 | **71.24** | **62.26** | **71.91** | **68.74** |
> | 0.4 | 71.15 | 62.04 | 71.37 | 68.19 |
> | 0.5 | 70.92 | 61.75 | 71.16 | 67.94 |
>
> **Q3**: Performance on multi-source DG for segmentation.
>
> **R**: Thanks for your valuable suggestion. We've accordingly added an experiment setting of domain generalized semantic segmentation, where two synthetic domains are jointly used for training. The attached results~show that, compared with the cases when simply using either one of the source domains, SBGen leads to further gains by pooling them together.
>
> | Method | Source Domain | ->C | ->B | ->M | Avg. |
> | ------ | ----- | ------- | ------ | ----- | ----- |
> | Baseline | G | 68.88 | 59.18 | 70.10 | 66.05 |
> | Baseline | S | 57.99 | 52.43 | 54.87 | 55.10 |
> | Baseline | G+S | **68.06** | **60.57** | **70.95** | **66.53** |
> | Ours | G | 71.24 | 62.26 | 71.91 | 68.74 |
> | Ours | S | 59.84 | 54.69 | 57.02 | 57.18 |
> | Ours | G+S | **72.81** | **64.15** | **73.28** | **70.08** |
>
> **Q4**: Ablation Study on Classification.
>
> **R**: Per your suggestion, we've accordingly added an ablation study on each component, tested on the classification benchmarks.
> As attached, each of the three components shows a similar contribution on classification accuracy.
>
> | DFS | SCE | SDI | PACS | VLCS | OfficeHome | DomainNet | TerraInc | Avg. |
> | ------ | ----- | ------- | ------ | ----- | ----- | ------ | ----- | ----- |
> | $\times$ | $\times$ | $\times$ | 95.5 | 82.4 | 86.5 | 59.6 | 58.3 | 76.5 |
> | $\checkmark$ | $\times$ | $\times$ | 96.3 | 83.9 | 87.4 | 61.8 | 59.5 | 77.8 |
> | $\checkmark$ | $\checkmark$ | $\times$ | 96.9 | 85.2 | 88.6 | 63.3 | 61.2 | 79.0 |
> | $\checkmark$ | $\times$ | $\checkmark$ | 97.1 | 86.0 | 89.5 | 63.9 | 62.6 | 79.8 |
> | $\checkmark$ | $\checkmark$ | $\checkmark$ | **97.4** | **86.7** | **89.9** | **64.4** | **63.5** | **80.4** |
>
> Should you have further comments or questions, we are happy to address during the author-reviewer discussion stage.

---

> > ### Comment · Reviewer_FGsB · 2025-08-06
> >
> > Thank you to the authors for the prompt response. I will maintain my score. My primary reason is that the contribution of this line of work to the community appears limited, particularly since many generalization issues can be addressed by scaling model size and training data. Moreover, it remains unclear whether the proposed method is applicable at larger scales.

---

> ### Author Response · Authors · 2025-08-07
> **Re: Official Comment by Reviewer FGsB**
>
> We are glad to see that the weakness \& questions in the review report have been addressed. We sincerely thank the reviewer for the continued evaluation and the time spent on our paper.
>
> We respectfully respond to the two new concerns raised in the final comment, which were not listed in the original weaknesses or clarification questions.
>
> **Q1**: The contribution of this line of work appears limited, as many generalization issues can be addressed by scaling model size \& training data.
>
> **R**: We appreciate the reviewer’s new concern, but respectfully disagree:
>
> - Model scaling and data scaling is an independent research line and could not degrade the contribution and significance of domain generalization. Indeed, larger models and data help, but they are not always available. One of the example is in the medical imaging or remote sensing scenario, where each center (domain) can only collect a small amount of images. Besides, due to the privacy issue and data management policy, the images from some centers (domains) may not be available. As a result, the generalization to unseen domains is still in great demand [1, 2]. Another fundemental issue regarding data scaling is the distribution shift. Different datasets can have different distribution, and always encounter with challenges such as long-tailed distribution [3] and out-of-distribution [4]. As a result, generalization still matters and plays an important role, for both machine learning and data scaling [5].
>
> As emphasized in recent machine learning and computer vision papers, domain generalization remains a crucial challenge under fixed resource and distribution shift settings.
>
> [1] Nguyen, A. Tuan, Philip Torr, and Ser Nam Lim. "Fedsr: A simple and effective domain generalization method for federated learning." Advances in Neural Information Processing Systems 35 (2022): 38831-38843.
>
> [2] Zhang, Ruipeng, et al. "Federated domain generalization with generalization adjustment." Proceedings of the IEEE/CVF Conference on Computer Vision and Pattern Recognition. 2023.
>
> [3] Shao, Jie, et al. "DiffuLT: Diffusion for Long-tail Recognition Without External Knowledge." Advances in Neural Information Processing Systems 37 (2024): 123007-123031.
>
> [4] Dong, Hao, et al. "Multiood: Scaling out-of-distribution detection for multiple modalities." Advances in Neural Information Processing Systems 37 (2024): 129250-129278.
>
> [5] Zhang, Boxuan, et al. "What if the input is expanded in OOD detection?." Advances in Neural Information Processing Systems 37 (2024): 21289-21329.
>
> - Our work provides a principled and lightweight solution, which has potential in the context of data scaling. Specifically, it formulates feature evolution via a Schrödinger Bridge, requiring no extra data and minimal tuning. This benefits the broader community by offering robust generalization under fixed foundation models, especially under the cases when re-training is infeasible.
>
> **Q2**: Whether the method is applicable at larger scales.
>
> **R**:  We appreciate the reviewer's continued evaluation on scalability. We are sorry that the scaling of training data in the rebuttal stage is presented in a way less clear than intended, where the performance on larger scale has already been validated. Specfically, the original experiment in the submission is conducted on GTA5 (G) dataset as the source domain, with 9.4K images. However, we further test the situation on:
>
> - SYNTHIA (S) dataset as the source domain, with $\sim$25K images.
>
> - GTA5(G) and SYNTHIA (S) datasets jointly as the source domains, with a total of $\sim$34.4K images.
>
> | Method | Source Domain | Data Size | Data Scale | ->C | ->B | ->M | Avg. |
> | ------ | ----- | ----- |  ----- | ------- | ------ | ----- | ----- |
> | Baseline | G | 9.4K| $\times$1 | 68.88 | 59.18 | 70.10 | 66.05 |
> | Baseline | S | 25K | $\times$2.5 | 57.99 | 52.43 | 54.87 | 55.10 |
> | Baseline | G+S | 34.4K| $\times$3.6 | 68.06 | 60.57 | 70.95 | 66.53 |
> | Ours | G | 9.4K| $\times$1 | 71.24 | 62.26 | 71.91 | 68.74 |
> | Ours | S | 25K | $\times$2.5 | 59.84 | 54.69 | 57.02 | 57.18 |
> | Ours | G+S | 34.4K| $\times$3.6 | 72.81 | 64.15 | 73.28 | 70.08 |
>
> We respectively raise the reviewer's attention that the scales of $\times$2.5 and $\times$3.6 have both been tested, where the proposed method still shows a clear improvement over the foundation model baseline. It indicates that its generalization ability maintains on two much larger data scales.
>
> Besides, technically, we would like to reminder the reviewer that the scaling ablity of the proposed method holds because:
>
> - It builds directly upon pre-trained large-scale VLMs (CLIP, EVA-02) and operates on top of them without changing the backbone, making it naturally compatible with scaling.
>
> - The proposed drift module is lightweight (an MLP over feature tokens) and SDE simulation is parallelizable. In practice, we apply it to multi-class semantic segmentation on full-resolution datasets with no performance bottlenecks.

---

### Official Review · Reviewer_HFke · 2025-07-02

**Clarity:** 3
**Significance:** 3
**Originality:** 3
**Rating:** 5
**Confidence:** 3

**Summary:**

The paper proposes a domain alignment framework based on Schrödinger Bridge methodologies, named SBGen, for visual domain generalization. By utilizing domain-invariant textual queries, the framework recognizes domain-aware features in the visual domain and, through a stochastic process in the feature space, transports them to domain-agnostic text embedding space. As a result, source domain features are transformed into generalized representations. The framework is supported by state-of-the-art performance on two different domain generalization tasks, namely image classification and segmentation.

**Questions:**

* Could the authors provide more information on which optimal transport solving methodology was used to model the domain alignment procedure?
* How costly is the bridge modeling for training the models? Please quantify total training time and FLOPs so the scalability of the framework can be gauged.

**Ethical Concerns:**

["NO or VERY MINOR ethics concerns only"]

**Final Justification:**

I will retain my rating of 5, given that the metholodogy has sufficient improvements on the baselines and also touches a novel idea in generalization and modeling. Authors have explained that target data is not used, which I think is significant as well.

**Limitations:**

Yes

**Quality:**

3

**Strengths And Weaknesses:**

Strengths:
* The idea of stochastically modeling the feature space between text and image is novel and provides sufficient space for more generalized representations to occur, thoroughly supported by the extensive experimentation.
* The paper is supported by sufficient mathematical explanation and each proposed module has a clear purpose into solving a particular problem within the task or within the framework. Ablation studies also support each module effectively and timestep ablation supports the role of the Schrödinger Bridge.

Weaknesses:
* Technical details on which methodology for Schrödinger Bridges is utilized for modeling the transport are missing. These types of problems typically require certain solving strategies, such as Conditional Flow Matching [1] or Iterative Proportinal Fitting [2], some details on which methods are utilized or if different methods have a different result would strengthen the work.

[1] Tong, Alexander, et al. "Simulation-free schr\" odinger bridges via score and flow matching." arXiv preprint arXiv:2307.03672 (2023).

[2] De Bortoli, Valentin, et al. "Diffusion schrödinger bridge with applications to score-based generative modeling." Advances in Neural Information Processing Systems 34 (2021): 17695-17709.

---

> ### Author Rebuttal · Authors · 2025-07-30
>
> We appreciate the reviewer for the positive comments on the technical novelties, extensive experiments and module design. The below is a point-by-point response to clarify the optimal transport method and the computational cost.
>
> **W1 \& Q1**: Details on the optimal transport solving method.
>
> **R**: Thanks for your insightful question. Concretely, our method is an entropy-regularized optimal transport method by solving the stochastic differential equation, which is different from existing methods. The steps are described as follows.
>
> First, we initialize the image feature $\boldsymbol{z}_0$ by the image encoder and the proposed Domain-aware Visual Feature Selection component, which is used as the starting point of the optimal transport.
>
> Then, the image feature $\boldsymbol{z}\_{t}$ of each step of the stochastic evolution is updated by the time embedding $\gamma(t)$ and the drift term, computed as
> $$ \text{drift} = \text{mlp}( \boldsymbol{z}\_{t} + \boldsymbol{q}\_{c} + \gamma(t))$$
> where $\boldsymbol{q}\_c$ is the text embedding, and $\text{mlp}$ denotes the drift network.
>
> Then, in every time step $t$, $\boldsymbol{z}\_{t}$ is updated by the stochastic differential equation.
> Specifically, based on the noise term (generated by a normal distribution $\mathcal{N}(0, I)$ to mimic the random permutation) and the drift term $\text{drift}$, it is computed as
> $$ z_{t+1} = z_{t} + \text{drift} \times dt + \sqrt{2 \epsilon dt} \times \mathcal{N}(0, I)$$
> where $dt$ is the time interval, and $\epsilon$ is a regularized parameter to control the intensity of the noise.
>
> Then, we compute the $K-L$ divergence of the drift noise's sum square, which will be later used as a regularization to smooth the transport path. It is computed as
> $$ \text{KL term} = \frac{ (\text{drift}^2) \cdot dt }{4 \epsilon}$$
>
> After $T$ time steps, the final image feature $z_T$ is aligned with the text embedding $\boldsymbol{q}_c$ by the mean square error (MSE) loss, given by
> $$ \text{MSE loss} = \text{MSE}( z_T, \boldsymbol{q}_c ) $$
>
> Theoretically, the final error regularized optimal transport (EROT) loss $\text{EROT loss}$ is a weighted sum of the MSE loss and the $K-L$ loss, given by
> $$ \text{EROT loss} = \text{MSE loss} + \epsilon \times \text{KL loss}, $$
> where the $K-L$ loss is an average of all the $T$ time steps, given by
> $$ \text{KL loss} = \frac{1}{T} \sum_{t=1}^{T} \text{KL term}_t $$
>
> Notice that, the computation of the $K-L$ loss is approximated in practical, where the approximation details are mentioned in L208-209 in the main text.
>
> We will enrich these details in the revised main text and supplementary material.
>
> To further check if different methods have a different result, we further compare the proposed method with the method in [1] and another commonly-used method, namely, Sinkhorn.
> The attached results show that these methods achieve a very similar result. The proposed method shows a slight improvement, which may be explained that it is more tailored for the alignment between image and text embeddings.
>
> | Method | G->C | G->B | G->M | Avg. | C->B | C->M | Avg. |
> | ------ | ----- | ------- | ------ | ------ | ----- | ------- | ------ |
> | Baseline | 68.88 | 59.18 | 70.10 | 66.05 | 64.72 | 76.15 | 70.44 |
> | [1] | 70.73 | 61.05 | 70.64 | 67.47 | 64.81 | 76.57 | 70.69 |
> | Sinkhorn | 69.62 | 60.18 | 70.57 | 66.79 | 65.29 | 76.48 | 70.89 |
> | Ours | **71.24** | **62.26** | **71.91** | **68.74** | **66.03** | **77.90** | **71.97** |
>
> We will add these results and discussions in the revised version.
>
> Since [2] does not have publicly available source code, we will cite and discuss it in the related work.
>
>
> **Q2**: Clarify the computational cost.
>
> **R:** Per your suggestion, we've accordingly compared the proposed method with the baseline in terms of the training time, GFLOPs and model size under the DGSS experimental setting.
> The attached table shows that although the proposed method achieves an acceptable trade-off between computational cost and performance improvement over the baseline.
> Specifically, the increase of GPU hour is 0.2 hours, the parameter number increase is 1.58 million, and the model size increase is 0.01GB. The GPU hour refers to the A100 GPU hardware.
>
> | Method | Training Time | Parameter Number | Model Size | GFLOPs | G->C | G->B | G->M | Avg. |
> | ------ | ----- | ------- | ------ | ------ | ----- | ------- | ------ | ----- |
> | Baseline | 79.0 GPU hours |	788.59M | 5.60GB | ~4.73 | 68.88 | 59.18 | 70.10 | 66.05 |
> | Ours | 79.2 GPU hours	| 790.17M | 5.61GB | ~4.74 | **71.24** | **62.26** | **71.91** | **68.74** |
>
> Should you have further comments or questions, we are happy to address during the author-reviewer discussion stage.

---

> > ### Comment · Reviewer_HFke · 2025-08-01
> > **Reponse to rebuttal**
> >
> > Thank you to the authors for addressing my concerns. I believe that the performance gain is substantial for the trade-off in complexity. I will retain my rating.

---

> > > ### Author Response · Authors · 2025-08-01
> > > **Re: Reponse to Reviewer HFke**
> > >
> > > Thanks for your swift response and the positive recommendation. We will incooperate all the suggestions when revising our manuscript.

---

### Note · Authors · 2025-08-12

We sincerely thank the Area Chair and all reviewers for their time, constructive feedback, and engagement throughout the review process. We are encouraged that most reviewers expressed strong positive recognition of our work’s **novelty, clear motivation, theoretical grounding, and solid empirical results**, with multiple reviewers maintaining high recommendations after the rebuttal.

During the rebuttal phase, we provided detailed clarifications and additional analyses to address all concerns raised. These included elaborating on the **Schrödinger Bridge solving methodology** (HFke), quantifying **computational cost** (HFke, WcC6), clarifying the choice and **sensitivity of hyperparameters** (FGsB), confirming the **experimental setting** (WcC6), **refining the notation and figures** (WcC6) for clarity, and discussing the stability and contribution of the **stochastic semantic evolution** (WcC6, 3EAi). We also considered all suggestions for **additional ablations and analyses** (FGsB, 3EAi), which will guide future improvements to this work.

We appreciate all reviewers’ insightful comments, which have helped strengthen the technical soundness, clarity, and broader applicability of our approach. We thank the AC for considering our responses and conclusions during the final decision process.

---

### Decision · Program_Chairs · 2025-09-17

**Decision:**

Accept (poster)

**Comment:**

The paper proposes an approach for visual domain generalization called learning cross-modal Schrodinger bridge. The idea is to leverage textual features to identify domain-specific features and then transport them to domain-agnostic text embedding space via a stochastic process. This provides source domain features that are generalizable.
Among the main strengths identified by the reviewers were:
- idea of stochastically transporting and modeling is interesting
- method is explained well with theoretical analyses and empirical validations
- novel idea of using Schrodinger bridges, not explored in DG literature before
- well-justified idea with clear experimental results

Some of the important concerns from the reviewers were:

- Missing technical details on exact methodology for Schrödinger Bridges
- non-trivial complexity of Schrodinger bridge learning
- issues with notations and main figure presentation
- limited contribution of the work, as many generalization issues can be handled with scaling data and/or model
- ablation study on replacing the dynamic Schrodinger component with static alignment one

In the post-rebuttal discussion phase, two reviewers remain satisfied with the author's responses. However, two reviewers stayed negative by mentioning aspects of data and model scaling and paper presentation issues. The authors do provided further experiments to address scaling issue and how to address the presentation issues. AC believes that the method is novel and is well-supported by both theoretical and experimental validation (as acknowledged by all reviewers), the remaining concerns are of minor nature and should not warrant rejection. The presentation issues are limited to use of a symbol at few places and clarification in a figure. Therefore, the decision is to accept the paper.